# BOOSTING ENTROPY WITH BELL BOX QUANTIZATION

**Ningfeng Yang**
University of British Columbia
nxyang@ece.ubc.ca

**Tor M. Aamodt**
University of British Columbia
aamodt@ece.ubc.ca

## ABSTRACT

Quantization-Aware Pre-Training (QAPT) is an effective technique to reduce the compute and memory overhead of Deep Neural Networks while improving their energy efficiency on edge devices. Existing QAPT methods produce models stored in compute-efficient data types (e.g. integers) that are not information theoretically optimal (ITO). On the other hand, existing ITO data types (e.g. Quantile/NormalFloat Quantization) are not compute-efficient. We propose BBQ, the first ITO quantization method that is also compute-efficient. BBQ builds on our key insight that since learning is domain-agnostic, the output of a quantizer does not need to reside in the same domain as its input. BBQ performs ITO quantization in its input domain, and returns its output in a compute-efficient domain where ITO data types are mapped to compute-efficient data types. Without sacrificing compute efficiency, BBQ outperforms prior SOTA QAPT methods by a perplexity reduction of up to 2 points for 4-bit models, up to 4 points for 3-bit models, up to 5 points for 2-bit models, and up to 18 points for 1-bit models. Code is available at https://github.com/1733116199/bbq.

## 1 INTRODUCTION

Quantization is an effective method to reduce the computation/memory/energy consumption of Deep Neural Networks (DNNs), allowing DNNs to be deployed to edge devices with limited hardware resources. However, quantization often degrades model quality. While Post-Training Quantization (PTQ) methods (Lin et al., 2024; Shao et al., 2024; Ma et al., 2024b; Liu et al., 2024a; Kim et al., 2024) can mitigate quality degradation without re-training for higher precisions (4+-bit), they struggle to maintain quality when weights and activations are quantized to 4-bit and below (Panferov et al., 2025; Esser et al., 2020; Ma et al., 2024a; Kumar et al., 2025). Quantization-Aware Training (QAT) methods (Panferov et al., 2025; Esser et al., 2020; Ma et al., 2024a; Kumar et al., 2025; Shen et al., 2024; Liu et al., 2025; Yamamoto, 2021), on the other hand, can achieve higher accuracy than PTQ methods under the same precision (Du et al., 2024; Panferov et al., 2025; Liu et al., 2024b) by introducing quantization in the training loop.

QAT can be further divided into Quantization-Aware Pre-Training (Panferov et al., 2025; Yang & Aamodt, 2025) (QAPT) and Quantization-Aware Fine-Tuning (Malinovskii et al., 2024; Du et al., 2024) (QAFT). QAFT initializes a low-precision model from a full-precision pre-trained checkpoint, and trains the model for a short duration to fit a downstream task, which is typically much smaller than the pre-training dataset (Du et al., 2024). On the other hand, QAPT initializes a low-precision model from scratch, aims to fit a much larger dataset, and typically trains for much longer durations. Compared to first pre-training in full-precision and subsequently applying PTQ/QAFT, QAPT may have higher pre-training speed (Kumar et al., 2025; Xi et al., 2023; Castro et al., 2025) as the forward pass of training may be performed in low precision.

This work aims to improve the accuracy of QAPT without compromising the resulting model's efficiency on edge devices. Specifically, we target the case when a low-precision model is initialized randomly, trained for long durations on large datasets, and is intended to be deployed to edge devices with constraints on memory capacity, inference latency, and energy consumption. The energy constraint requires the model to be compute-efficient, or that expensive high-precision matrix multiplications be substituted with low-precision arithmetic, while the memory and latency constraints require the model to be small in size, or to have a low parameter count and precision. A main chal-

| Signed INT4 | -8, $\pm 7$, $\pm 6$, $\pm 5$, $\pm 4$, $\pm 3$, $\pm 2$, $\pm 1$, 0 |
|---|---|
| MX FP4 | $\pm 6$, $\pm 4$, $\pm 3$, $\pm 2$, $\pm 1.5$, $\pm 1$, $\pm 0.5$, $\pm 0$ |

Table 1: Possible values of the INT4 and the MX FP4 data type.

lenge of QAPT is that models with limited memory footprint cannot fit large datasets well due to a lack of learning capacity (Kumar et al., 2025).

Using the Shannon entropy (Shannon, 1948; Cover & Thomas, 2006) of quantized weights as a proxy for the amount of information/knowledge present in a model trained with QAPT, we observe that SOTA QAPT methods QuEST (Panferov et al., 2025) and LSQ (Esser et al., 2020) produce quantized models that under-utilize the available learning capacity. The under-utilization is because LSQ and QuEST use compute-efficient data types (e.g. integers and floats) which are not information theoretically optimal (ITO). While existing ITO data types (Dettmers et al., 2023; 2022) can reduce this under-utilization of learning capacity, they lack compute efficiency on modern CPUs/GPUs, which limits their applicability on edge devices with limited energy.

To maximally utilize the limited learning capacity while preserving compute efficiency, we propose Bell Box Quantization (BBQ). BBQ is designed based on our key insight: since learning is domain-agnostic, the output of a quantizer does not need to live in the same domain as its input. BBQ performs ITO quantization in its input domain to maximally preserve information, but returns its output in a different domain, where ITO data types are mapped to compute-efficient data types. BBQ achieves higher capacity utilization and better prediction quality than QuEST and LSQ. Unlike existing ITO data types, a BBQ-quantized model can still accelerate expensive matrix multiplications with low-precision arithmetic.

## 2 BACKGROUND AND MOTIVATION

In this section, we discuss compute-efficient data types on modern GPUs, existing ITO quantization methods, and lastly, the domain-agnostic property of learning which is the key inspiration of BBQ.

Quantization is the discretization of a continuous random[1] variable $x$ to a discrete random variable $\hat{x}$, such that $\hat{x}$ is sufficiently close to $x$. We present a generic template for quantization as follows:

$$\hat{x} = f^{-1}(q), \text{ and } q = r(f(x)) \tag{1}$$

where $f : \mathbb{R}^d \to \mathbb{R}^d$ is a transform function that converts $x \in \mathbb{R}^d$ to an intermediate domain suitable for quantization; $r : \mathbb{R}^d \to \mathbb{T}^d$ is a rounding function that takes real-valued inputs and returns outputs $q$ in a set of $2^b$ real values, i.e. $q \in \mathbb{T}^d$, $\mathbb{T} \subseteq \mathbb{R}$, and $|\mathbb{T}| = 2^b$; and $f^{-1}$, the inverse function of $f$, is responsible for converting the rounded variable $q$ back to the original domain of $x$. Throughout this work, $x$, $q$, and $\hat{x}$ represent neural network weights or activations that are **raw**, **quantized**, and **de-quantized**, respectively.

### 2.1 COMPUTE EFFICIENT DATA TYPES

A data type $\mathbb{T}$ is compute-efficient if hardware offers low-precision multiply-accumulate instructions that directly operate on members of $\mathbb{T}$ without first decoding to high-precision data types. Table 1 shows members of $\mathbb{T}$ for INT4, which is supported by the Nvidia Ampere (Nvidia, 2021) and Turing (Nvidia, 2018) architectures, and members of $\mathbb{T}$ for MX FP4 (Rouhani et al., 2023), which is supported by the Blackwell architecture (Nvidia, 2025). When $f^{-1}$ is linear or affine, a compute-efficient data type can be used to accelerate matrix multiplication and convolution (Yao et al., 2021). For example, if $f^{-1}(q) = sq$ for a constant scalar $s$, then the multiplication of activation matrix $\hat{X} = f^{-1}(Q_x) = sQ_x$ and weight matrix $\hat{W} = f^{-1}(Q_w) = sQ_w$ can be simplified to $s^2(Q_x \cdot Q_w)$. In other words, the matrix multiplication can be computed using solely low-precision representations of activations and weights, and later de-quantized to the original domain by multiplication of $s^2$.

Prior SOTA QAPT methods QuEST (Panferov et al., 2025) and LSQ (Esser et al., 2020) use linear/affine $f^{-1}$ and compute-efficient data types. We show the definition of $f$, $r$, and $f^{-1}$ for LSQ

---

[1]We assume $x$ is a random variable, or that $x$ is sampled from some distribution, to enable the use of statistical properties like the Shannon entropy.

and QuEST as follows:

$$\text{LSQ: } f(x) = x/s, r(v) = \lfloor \text{clip}(v, -2^{b-1}, 2^{b-1} - 1) \rceil, f^{-1}(q) = sq$$

$$\text{QuEST: } f(x) = \frac{\text{HT}(x)}{\alpha^* \sigma} - \frac{1}{2}, r(v) = \lfloor \text{clip}(v, -2^{b-1}, 2^{b-1} - 1) \rceil, f^{-1}(q) = \text{HT}(\alpha^* \sigma(q + \frac{1}{2})) \tag{2}$$

where $s$, $\sigma$, and $\alpha^*$ are scalars; all division operations are element-wise; HT is the Hadamard transform, a linear operation; and the clip operation and the round-to-nearest-integer operation $\lfloor \cdot \rceil$ enforce the output of $r$ to be a $b$-bit signed integer. Since $f^{-1}$ is linear or affine in LSQ and QuEST, matrix multiplications can be accelerated with low-precision arithmetic.

In short, a compute-efficient data type $\mathbb{T}$ and a linear/affine dequantization function $f^{-1}$ are two key features that enable quantization methods like LSQ and QuEST to run efficiently on modern GPUs.

## 2.2 Information Theoretically Optimal Quantization Methods

An ITO quantization method is one that ensures each member of $\mathbb{T}$ is used equally often (Dettmers et al., 2023). For example, if $b = 2$, then 25% of elements of $x$ should be assigned to each of the 4 possible values in $\mathbb{T}$. While there are many possible mappings that evenly divide the elements of $x$, a trivial mapping is to use the 25th, the 50th, and the 75th percentiles as rounding boundaries. In other words, elements of $x$ less than the 25th percentile of $x$ are rounded to the smallest value in $\mathbb{T}$; elements larger than the 25th percentile but less than the 50th percentile are rounded to the second smallest value in $\mathbb{T}$, etc. If all $d$ elements of $x$ are i.i.d. sampled from $N(0, \sigma^2)$ and $d$ is sufficiently large, this trivial mapping can be expressed as:

$$f(x) = x/\sigma, r(v) = T[\text{floor}(2^b \Phi(v))], f^{-1}(q) = \sigma q \tag{3}$$

where $T[i]$ is the $i$th smallest element in $\mathbb{T}$. Equation 3 only requires that $\mathbb{T}$ contains $2^b$ unique real values, regardless of what these real values may be. As such, Equation 3 describes a set of ITO methods rather than a unique one. By evenly dividing the elements of $x$, all ITO data types maximize the empirical Shannon entropy of $q$ defined as

$$H(q) = \sum_{t \in \mathbb{T}} -P(q, t) \log_2 P(q, t) \tag{4}$$

where $P(q, t)$ is the probability that a (uniformly) randomly chosen element of $q$ is equal to $t$. Note that entropy was shown to positively correlate with model accuracy/prediction quality (Cheng et al., 2025; Shen et al., 2024).

Inspired by ITO quantization methods, Dettmers et al. (2023) proposed NF4, a 4-bit NormalFloat whose values are a list of abs-max-normalized Gaussian quantiles (Yoshida, 2023; Dotzel et al., 2024). Due to lack of hardware support, NF4 must be de-quantized to full-precision floats and then used in computation, which limits its applicability on energy-constrained edge devices.

This presents a dilemma: while ITO quantization methods maximally preserve information, they are not compute-efficient; compute-efficient quantization methods, on the other hand, are not ITO for Gaussianly distributed weights/activations. Can we obtain the best of both worlds?

## 2.3 Learning is Domain-Agnostic

Since neural networks are universal function approximators (Hornik et al., 1989), they are capable of learning from transformed/augmented data, or even data encoded in latent spaces that are not human-understandable. To list a few examples: DNNs can correctly classify images that are rotated (He et al., 2015); DNNs can classify images that are transformed to the frequency domain (Xu et al., 2020); When autoencoders (Gao et al., 2025) are trained just to reconstruct the data, the resulting latent embedding of the data can be used by other DNNs to complete downstream tasks. These examples are all evidence that, as long as information is preserved, simply projecting/transforming data to a different domain does not prevent learning.

The domain-agnostic property of learning provides an opportunity to circumvent the inefficiency of ITO quantization methods. Rather than treating quantizers as data compressors and reconstructors,

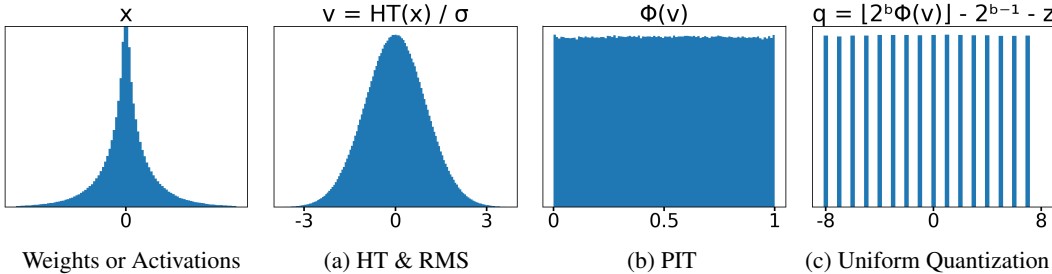

Figure 1: The three steps of the BBQ quantization formula (Equation 5) for $b = 4$. Step a is the Hadamard Transform (HT) followed by RMS Normalization. Step b is the probability integral transform (PIT). Step c is uniform quantization. We name our method Bell Box Quantization because Figure 1a looks like a bell and Figure 1b looks like a box (rectangle) which is quantized in Figure 1c.

we could treat them as feature extractors that return a set of compact latent features of the original data. Critically, these latent features reside in an output domain, that is, not necessarily the same domain as the input, but is designed to be a compute-efficient domain, where features can be efficiently used by matrix multiplication operators. While N2UQ (Liu et al., 2022) also returns outputs in a compute-efficient but distinct-from-input domain, BBQ is the first method to apply such a technique to ITO quantization for both activations and weights. In addition, N2UQ assumes raw weights are uniformly distributed, while BBQ is ITO without making any assumptions on the distribution shape of raw weights or raw activations. In Section 4, we quantitatively compare BBQ against N2UQ, showing BBQ achieves lower perplexity under the same precision. Inspired by the domain-agnostic property of learning, we ask the research question: *During QAPT, can ITO quantization outperform non-ITO methods if we return outputs in a compute-efficient domain?*

## 3 BELL BOX QUANTIZATION

In this section, we present the Bell Box Quantizer (BBQ). BBQ is designed to be ITO, or to maximally preserve information from its input $x$ (activations or weights), while also returning compute-efficient outputs that can be accelerated by modern hardware. The BBQ quantization and dequantization formulas are shown in Equations 5 and 6, respectively:

$$q = \lfloor 2^b \Phi(v) \rfloor - 2^{b-1} - z, \text{ where } v = \text{HT}(x)/\sigma \tag{5}$$

$$\hat{x} = \frac{\gamma}{2^{b-1}} q \tag{6}$$

where HT : $\mathbb{R}^d \rightarrow \mathbb{R}^d$ is the Hadamard transform operation; the root-mean-square (RMS) normalization factor, $\sigma$, is defined as $\sigma = (1/d \cdot \sum_{i=1}^{d} (\text{HT}(x)_i)^2)^{1/2}$; $v$ is a variable introduced for notational convenience; $\Phi : \mathbb{R}^d \rightarrow (0, 1)^d$ is the element-wise Gaussian CDF operation; $\lfloor \cdot \rfloor$ is the element-wise floor operation; $z \in \{0, -0.5\}$ is a hyperparameter zero point; and $\gamma$ is a learnable scaling factor. Figure 1 illustrates the three major steps of Equation 5: Hadamard Transform combined with RMS Normalization (step a), probability integral transform (step b), and uniform quantization (step c). We discuss steps a, b, and c in Sections 3.1, 3.2, and 3.3, respectively, and the dequantization formula Equation 6 in Section 3.4.

### 3.1 STEP A: HADAMARD TRANSFORM AND RMS NORMALIZATION (FIGURE 1A)

Since ITO quantization is ill-defined without knowing a distribution, the first step of BBQ is to convert its input $x$ from an **unknown** distribution to a Gaussian by applying the Hadamard Transform (Panferov et al., 2025; Yang et al., 2025). After the transform, we assume elements of $\text{HT}(x)$ behave like samples from $N(0, \sigma^2)$ following Panferov et al. (2025) and Yang et al. (2025). Instead of transforming all elements of $x$, we follow QuEST and perform the Hadamard Transform on every $H$ elements of $x$ along the input channel dimension. During training, the Hadamard Transform is a differentiable operation, and we simply use autograd frameworks to perform its backward pass. During inference, the Hadamard transform is implemented with a vector-matrix multiplication between an $H$-element slice of $x$ and the pre-computed $H \times H$ Hadamard matrix.

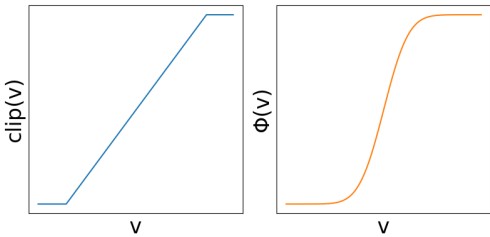 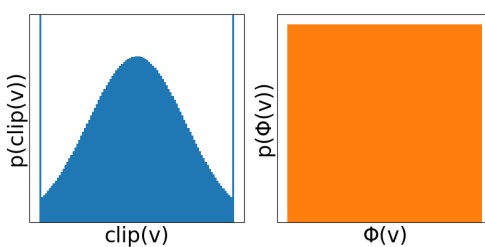

(a) Graph of clip and $\Phi$. Both functions take input in $(-\infty, \infty)$ and return output in a finite range, and therefore both qualify as clipping functions.

(b) PDF of clip$(v)$ and $\Phi(v)$ for $v \sim N(0,1)$. $\Phi(v)$ has maximum (differential) entropy as its probability mass is evenly spread out.

Figure 2: Comparison of clip (blue) and the standard Gaussian CDF $\Phi$ (orange).

Next, we apply RMS normalization by multiplying the Gaussian-like data by $1/\sigma$. As a result, elements of $v = \text{HT}(x)/\sigma$ behave like samples from $N(0,1)$. During training, we use autograd frameworks to perform the backward pass of RMS Normalization. Following standard quantization practice (Nagel et al., 2021), we perform channel-wise RMS normalization for weights, and per-tensor RMS normalization for activations.

We discuss an optional variant of BBQ, named BBQ-Fast, which achieves identical perplexity (Section A.3) but has better inference speed. During training, BBQ-Fast has identical behaviour to BBQ, except BBQ-Fast additionally keeps track of the exponential moving average of $1/\sigma$, denoted as $E_{1/\sigma}$, using the update rule in Equation 7, which is executed every training iteration with $\beta = 0.99$.

$$E_{1/\sigma} \leftarrow \beta \cdot E_{1/\sigma} + (1 - \beta) \cdot (1/\sigma) \tag{7}$$

During inference, BBQ-Fast performs activation RMS normalization by multiplying $\text{HT}(x)$ by the final value of $E_{1/\sigma}$, instead of by $1/\sigma$, since measuring $\sigma$ requires the execution of the root-mean-square operation, which often incurs expensive cross-thread-block communication for large activation tensors.

## 3.2 STEP B: PROBABILITY INTEGRAL TRANSFORM (FIGURE 1B)

The probability integral transform (David & Johnson, 1948; Fisher, 1932) converts any continuous distribution to a uniform one by applying its CDF on the data. We apply the standard Gaussian CDF $\Phi$ to $v$ which behaves like samples from the standard Gaussian $N(0,1)$, creating $\Phi(v)$ which behaves like samples from $U(0,1)$. Function $\Phi$ is designed to clip data within a finite range (Figure 2a) while maximizing entropy (Figure 2b) for Gaussianly distributed data, and is meant to replace the vanilla clip function in QuEST and LSQ. Figure 2a shows that $\Phi$ is much smoother than clip. This is because $\Phi$ is infinitely differentiable and clip is piecewise linear. A smoother operation can be more suitable (Nesterov, 2014; Bottou et al., 2018) for optimization methods like Gradient Descent. For example, the GELU function (Hendrycks & Gimpel, 2023) defined as $\text{GELU}(x) = x \cdot \Phi(x)$ is smoother than the piecewise linear ReLU (Agarap, 2018), and GELU can outperform ReLU empirically (Hendrycks & Gimpel, 2023). During training, $\Phi$ is a differentiable operation and we use the autograd framework to perform BackProp. During inference, we combine $\Phi$ with the subsequent floor operation using the following property, which is true $\forall v \in \mathbb{R}, \forall i \in \{0, 1, 2, \dots 2^b - 1\}$:

$$\lfloor 2^b \Phi(v) \rfloor = i \iff \Phi^{-1}\left(\frac{i}{2^b}\right) \leq v < \Phi^{-1}\left(\frac{i+1}{2^b}\right) \tag{8}$$

Property 8 allows us to pre-compute $\Phi^{-1}(i/2^b)$ for all $2^b$ possible values of $i$, and at runtime perform a binary search that requires $b$ floating point comparisons. Our profiling results in Section 4.2 show this binary search incurs negligible runtime overhead. The pseudocode of a 3-bit inference kernel implementation of BBQ is shown in Algorithm 1.

## 3.3 STEP C: UNIFORM QUANTIZATION (FIGURE 1C)

After applying the Probability Integral Transform, since $\Phi(v)$ behaves like samples from $U(0,1)$, we can achieve ITO quantization by applying uniform quantization, obtaining $\lfloor 2^b \Phi(v) \rfloor$ which can

---

**Algorithm 1** Example 3-bit Inference Quantization Kernel for a Thread Block

---

1: **function** BBQ(block_id, x_ptr, h_ptr, q_ptr, $E_{1/\sigma}$)
2:     x = x_ptr[block_id:block_id+1, :] # x_ptr is an $M \times N$ matrix reshaped to $(MN/H, H)$
3:     h = h_ptr[:, :] # h_ptr is a pre-computed Hadamard matrix with shape $(H, H)$
4:     xh = x @ h # Step a: vector-matrix multiplication between shapes $(1, H)$ and $(H, H)$
5:     v = xh $\cdot E_{1/\sigma}$ # Step a: element-wise multiplication by pre-computed scalar (BBQ-Fast)
6:     q = an array with $H$ INT4/MX FP4 elements
7:     **for** $t \leftarrow 0$ **to** $H - 1$ **do** # can be done in parallel across warps and threads
8:         **if** v[t] $\geq \Phi^{-1}(4/8)$ **then** # Step b: Binary search with pre-computed $\Phi^{-1}$ values
9:             **if** v[t] $\geq \Phi^{-1}(6/8)$ **then**
10:                 q[t] = 3 if v[t] $\geq \Phi^{-1}(7/8)$ else 2 # Step c: quantize to INT4/MX FP4
11:             **else**
12:                 q[t] = 1 if v[t] $\geq \Phi^{-1}(5/8)$ else 0 # Step c: quantize to INT4/MX FP4
13:             **end if**
14:         **else**
15:             **if** v[t] $\geq \Phi^{-1}(2/8)$ **then**
16:                 q[t] = -1 if v[t] $\geq \Phi^{-1}(3/8)$ else -2 # Step c: quantize to INT4/MX FP4
17:             **else**
18:                 q[t] = -3 if v[t] $\geq \Phi^{-1}(1/8)$ else -4 # Step c: quantize to INT4/MX FP4
19:             **end if**
20:         **end if**
21:     **end for**
22:     q_ptr[block_id:block_id+1, :] = q # store quantized INT4/MX FP4 results to global memory
23: **end function**

---

| Precision $b$ | Zero Point $z$ | Possible Values of $q$ | INT4 | MX FP4 |
|---|---|---|---|---|
| 4 | 0 | -8, -7, -6, -5, -4, -3, -2, -1, 0, 1, 2, 3, 4, 5, 6, 7 | ✓ | ✗ |
| 3 | 0 | -4, -3, -2, -1, 0, 1, 2, 3 | ✓ | ✓ |
| 2 | -0.5 | -1.5, -0.5, 0.5, 1.5 | ✗ | ✓ |
| 1 | -0.5 | -0.5, 0.5 | ✗ | ✓ |

Table 2: The BBQ data type: zero point $z$, possible values of $q$, and whether BBQ can be encoded as INT4 and MX FP4 for each precision $b \in \{1, 2, 3, 4\}$.

be stored as a $b$-bit unsigned integer. Since zero-mean activations can contribute to faster convergence (Bjorck et al., 2018; Ioffe & Szegedy, 2015), we convert the non-negative data $\lfloor 2^b \Phi(v) \rfloor$ to symmetric data by subtracting $2^{b-1}$ and the hyperparameter zero point $z$, producing the final $q = \lfloor 2^b \Phi(v) \rfloor - 2^{b-1} - z$ stored as a signed compute-efficient data type. Inspired by NF4 (Dettmers et al., 2023), we use $z = 0$ for $b \in \{3, 4\}$, allowing the data type to represent zero exactly while sacrificing a value on the positive side. We use $z = -0.5$ for $b \in \{1, 2\}$ to ensure activations remain zero-mean. Table 2 lists the values supported by the BBQ data type. For $b \in \{1, 2, 3\}$, BBQ can be represented using the MX FP4 data type on existing Blackwell GPUs. For $b \in \{3, 4\}$, BBQ can be represented using INT4 on existing Ampere and Turing GPUs. During training, we use the Straight-Through Estimator (Bengio et al., 2013) for the floor operation, and the autograd framework for all other differentiable operations.

## 3.4 DEQUANTIZATION

Inspired by LSQ, the dequantization formula of BBQ, Equation 6, includes a learnable scaling parameter $s = \gamma/2^{b-1}$. A learnable scaling factor enables control of the magnitude of $\hat{x}$. To ensure the learnable scaling factor can be initialized to the same value regardless of precision $b$, we do not use $s$ directly but instead decouple $s$ into the ratio of a precision-independent learnable scaling factor $\gamma$ and a constant precision-dependent scaling factor $2^{b-1}$. Following LSQ, we apply gradient scaling (Esser et al., 2020) to reduce the gradient of $\gamma$ by a factor of $\sqrt{d}$, where $d$ is the number of weights/activations in $x$. Following Nagel et al. (2021), we use channel-wise $\gamma$ for weights and per-tensor $\gamma$ for activations.

| Params | Tokens | Full | Bits | BBQ | | QuEST | | LSQ | |
|---|---|---|---|---|---|---|---|---|---|
| | | | | Entropy | Perplexity | Entropy | Perplexity | Entropy | Perplexity |
| 95M | 3B | **24.75** | 4 | **3.93** | **25.51** | 3.61 | 26.37 | 3.59 | 27.46 |
| 95M | 3B | **24.75** | 3 | **2.96** | **26.55** | 2.78 | 29.04 | 2.74 | 30.27 |
| 95M | 3B | **24.75** | 2 | **1.97** | **31.34** | 1.92 | 35.58 | 1.69 | 36.58 |
| 95M | 3B | **24.75** | 1 | **1.00** | **49.22** | **1.00** | 67.78 | - | - |
| 125M | 5B | **21.51** | 4 | **3.93** | **22.15** | 3.61 | 22.98 | 3.60 | 23.77 |
| 125M | 5B | **21.51** | 3 | **2.96** | **23.22** | 2.78 | 25.21 | 2.74 | 26.28 |
| 125M | 5B | **21.51** | 2 | **1.98** | **27.34** | 1.93 | 31.32 | 1.81 | 31.42 |
| 125M | 5B | **21.51** | 1 | **1.00** | **44.58** | **1.00** | 72.54 | - | - |
| 200M | 10B | **17.93** | 4 | **3.93** | **18.79** | 3.61 | 19.06 | 2.73 | 1778 |
| 200M | 10B | **17.93** | 3 | **2.96** | **19.74** | 2.78 | 20.82 | 2.50 | 140.9 |
| 200M | 10B | **17.93** | 2 | **1.98** | **23.08** | 1.93 | 25.46 | 1.63 | 78.19 |
| 200M | 10B | **17.93** | 1 | **1.00** | **38.27** | **1.00** | 52.37 | - | - |
| 300M | 20B | **15.43** | 4 | **3.93** | **16.10** | 3.61 | 16.26 | - | - |
| 300M | 20B | **15.43** | 3 | **2.96** | **16.90** | 2.78 | 17.67 | - | - |
| 300M | 20B | **15.43** | 2 | **1.98** | **19.75** | 1.93 | 21.53 | - | - |

Table 3: Entropy and Perplexity of LLaMA models pre-trained on the C4 dataset. The headers are the number of parameters (Params), the number of tokens (Tokens), perplexity without any quantization (Full), activation/weight precision (Bits), and the entropy and perplexity of BBQ, QuEST, and LSQ.

The initialization of $\gamma$ plays a critical role, as overly large initialization can lead to gradient explosion while overly small initialization can lead to gradient vanishing. We initialize $\gamma$ to $\zeta^* \sigma_0$, where $\sigma_0$ is the $\sigma$ measured at the first iteration of training, and $\zeta^*$ is defined as:

$$\zeta^* = \arg \min_{\zeta} E_{v \sim N(0,1)}[(v - \zeta(2\Phi(v) - 1))^2] \tag{9}$$

See Section A.2 for a detailed explanation of $\zeta^*$ and Equation 9. In short, by initializing $\gamma$ to $\zeta^* \sigma_0$, $\hat{x}$ can have approximately the same magnitude as $x$ (in the first iteration) to prevent the activation magnitude from exploding or vanishing as tokens travel deeper into the network. After the first iteration, we simply let $\gamma$ update itself based on gradient descent. We do not apply weight decay to $\gamma$, as an overly decayed $\gamma$ may lead to gradient vanishing.

## 4 EVALUATION AND DISCUSSION

We run experiments to evaluate the perplexity of BBQ against other SOTA QAPT methods on representative LLM architectures. We use the publicly available source code of QuEST (Panferov et al., 2025), add the implementation of BBQ, and train on LLaMA (Touvron et al., 2023; Vaswani et al., 2017) models with $n$ non-embedding parameters plus $e$ embedding parameters, where $n \in \{30M, 50M, 100M, 200M\}$ and $e \in \{65M, 75M, 100M, 100M\}$. For each model, we pre-train with $100n$ C4 tokens while quantizing the weights and activations of all linear layers to $b$-bit following QuEST. We compare BBQ against QuEST/LSQ for each $b \in \{1, 2, 3, 4\}$, and present the evaluation quality (perplexity) and weight entropy[2] (bits) in Table 3. The omitted entries in Table 3 are because LSQ does not support 1-bit quantization and LSQ diverges for larger models (unlike BBQ which works well for 1-bit and converges even for larger models). We show zero-shot results in Table 9, and training loss curves in Section A.7. Each experiment for $n + e \leq 200M$ is conducted on one Nvidia RTX 5090 and lasts for up to 1 day. Each experiment for $n + e > 200M$ is conducted on one Nvidia A100 80GB and lasts for 3.5 days. In total, all experiments took approximately 1.5 GPU months. Lastly, we evaluate BBQ against QuEST, LSQ, ParetoQ-SEQ (Liu et al., 2025), N2UQ (Liu et al., 2022), and NormalFloat (Dettmers et al., 2023) on GPT (Brown et al., 2020) models and present our results in Table 4.

Results in Table 3 and 4 suggest BBQ can consistently achieve **higher entropy** and **lower perplexity** than QuEST and LSQ at the same precision, and that entropy can be a good proxy to prediction quality. A special case is when $b = 1$. In this case, both BBQ and QuEST achieve the maximal

---

[2]Note that when weights are quantized to $b$-bit, the maximum achievable weight entropy is $b$ bits. In other words, entropy is upper bounded by precision.

| Params | 95M | 95M | 95M | 95M | 95M | 95M | 95M | 95M | 95M | 95M | 95M | 95M |
|--------|------|------|------|------|------|-------|------|-------|-------|-------|-------|-------|
| Bits WA | 16 | 4 | 4 | 4 | 4 | 3 | 3 | 3 | 3 | 3 | 1 | 1 |
| Method | - | BBQ | QuEST | LSQ | NF4 | BBQ | QuEST | LSQ | NF3 | N2UQ | BBQ | QuEST |
| Perplex | 25 | **26.7** | 26.83 | 27.55 | 28.5 | **28.61** | 29.7 | 30.85 | 36.16 | 29.87 | **53.8** | 77.96 |
| Params | 95M | 95M | 95M | 95M | 95M | 95M | 125M | 125M | 125M | 125M | 125M | 125M |
| Bits WA | 2 | 2 | 2 | 2 | 2 | 2 | 3 | 3 | 2 | 2 | 1 | 1 |
| Method | BBQ | QuEST | LSQ | NF2 | SEQ | N2UQ | BBQ | QuEST | BBQ | QuEST | BBQ | QuEST |
| Perplex | **34.36** | 36.7 | 38.98 | 246.6 | 40.07 | 35.59 | **24.78** | 26.05 | **29.38** | 31.47 | **50.46** | 69.78 |

Table 4: Perplexity of GPT models pre-trained on the C4 dataset.

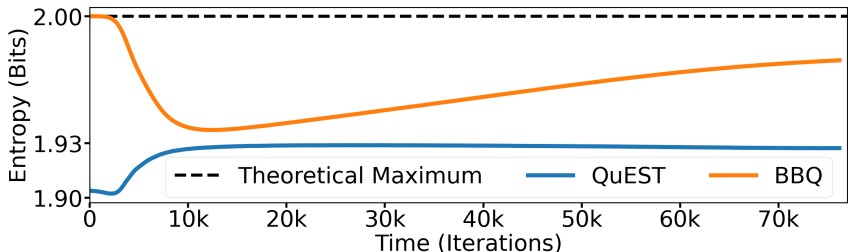

Figure 3: Quantized weight entropy vs. training iterations for LLaMA-300M with 2-bit weight and activations, pre-trained on 20 billion C4 tokens (batched into 80 thousand training iterations).

entropy of 1 bit. We attribute the performance gain of BBQ in the 1-bit case to the fact that $\Phi$ from BBQ is smoother than the clip operation from QuEST (as discussed in Section 3.2). In addition, we note that LSQ diverges for LLaMA-200M, which is reflected in its entropy. Notably, when LSQ does not diverge, like in the case of LLaMA-95M and LLaMA-125M, we see that LSQ can achieve an entropy of 3.6 bits for 4-bit precision, 2.74 bits for 3-bit precision, and 1.7 bits for 2-bit precision. However, when LSQ diverges, like in the case of LLaMA-200M, we see that its entropy drops to 2.73 bits for 4-bit precision, 2.50 bits for 3-bit precision, and 1.63 bits for 2-bit precision. This shows that entropy is a good indicator of a quantized model's prediction quality.

## 4.1 ENTROPY AND LEARNING CAPACITY

In this section, we discuss, based on empirical measurements of entropy during training, the improvement of BBQ over QuEST in terms of learning capacity utilization. Figure 3 shows the behaviour of weight entropy of BBQ and QuEST during 2-bit training. We note a few observations. First, BBQ can maximize entropy as it can achieve the theoretical maximum entropy of 2 bits at the beginning of training. Second, during training, a BBQ-quantized model can adjust (increase or decrease) its entropy. Third, excluding the first 8 thousand iterations when learning rate is warming up, BBQ tends to increase weight entropy for the last 72 thousand iterations. This suggests, as more training tokens are presented to the model, more information is accumulated in the weights. Lastly, the entropy of QuEST seems to have an empirical ceiling at around 1.93 bits. We attribute the empirical entropy ceiling of QuEST to the following: QuEST applies uniform quantization to $\mathrm{HT}(x)/\sigma$, and therefore QuEST is only ITO if $\mathrm{HT}(x)$ follows a **uniform distribution**, which is **unlikely** as the Hadamard transform has a tendency to **Gaussianize** (Yang et al., 2025; Panferov et al., 2025). See Section A.5 for a visualization of the difference between BBQ and QuEST.

By having a lower empirical entropy upper bound of 1.93 bits, QuEST limits the learning capacity of the model, while BBQ does not suffer from such capacity under-utilization. Suppose a model has $d$ parameters, each quantized to $b$ bits. This means the model has a total of $2^{db}$ possible unique states. Some of the $2^{db}$ states have higher entropy, while others have lower entropy. QAPT's objective is to find a (locally) optimal state out of all $2^{db}$ possible states. However, if QuEST enforces its weight entropy to be lower than 1.93 bits, then some of the high-entropy states cannot be explored by the training algorithm. Therefore, the model's learning capacity is effectively reduced.

## 4.2 INFERENCE SPEEDUP

In this section, we discuss our profiling results of the inference latency of BBQ, compared against FP16 and NF4. We perform our latency measurements on NVIDIA RTX 5090, a Blackwell GPU that

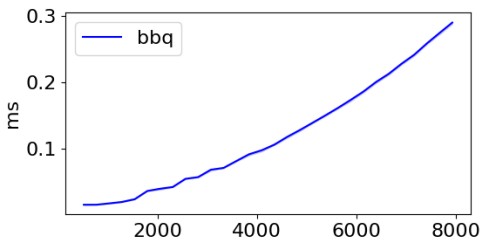 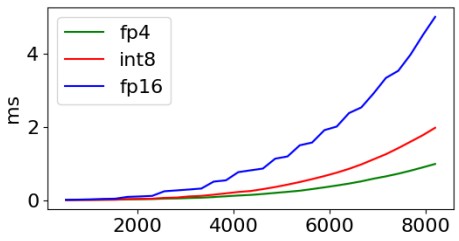

(a) BBQ quantization kernel of an $N \times N$ matrix  (b) Matrix multiplication of two $N \times N$ matrices

Figure 4: Kernel latency on Nvidia RTX 5090 (y-axis) vs. matrix size $N$ (x-axis).

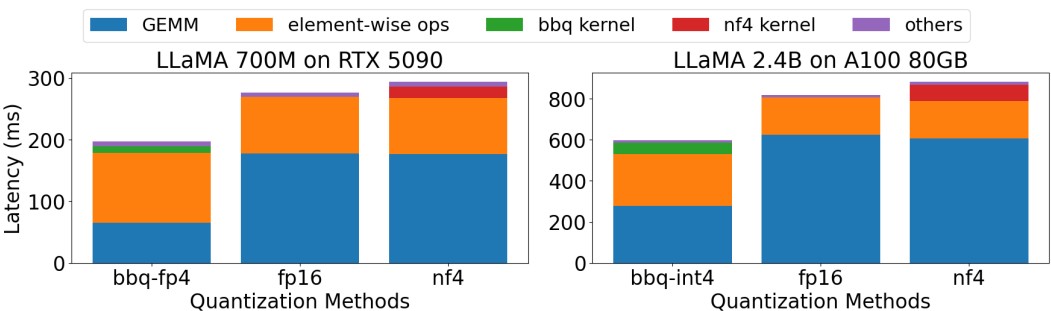

Figure 5: End-to-end LLaMA inference latency of BBQ, FP16 and NF4 on RTX 5090 and A100 GPUs. For each linear layer, BBQ launches an activation quantization kernel (green region), an fp4/int4 matrix multiplication kernel (part of blue regions), and an element-wise scaling kernel (part of orange region).

supports MX FP4 matrix multiplication, and NVIDIA A100 80GB, an Ampere GPU that supports INT4 matrix multiplication. At inference time, for each linear layer, BBQ launches an activation quantization kernel to compute the quantized activation matrix $Q_x$, a low-precision matrix multiplication kernel between $Q_x$ and the quantized weight matrix $Q_w$, and lastly an element-wise scaling kernel to scale the result $Q_x Q_w$ by a factor of $s_x s_w$. We note that BBQ performs its weight quantization **offline** and therefore weight quantization does **not** incur inference latency overhead. Since compute-efficiency, the feature that distinguishes BBQ from NF4, manifests in latency savings during the execution of compute-bound kernels, we focus on profiling the **prefill** phase of inference. However, we note both BBQ and NF4 can achieve latency reduction during the memory-bound **decode** phase as both can reduce memory bandwidth consumption. In addition, during the decode phase, BBQ may save **energy** compared to NF4 as **computation** is performed in low precision.

We first show at the kernel level, the latency of the BBQ activation quantization kernel is negligible compared to the time saved by performing matrix multiplication in low-precision. Figure 4a shows the latency of a Triton (Tillet et al., 2019) implementation (shown in Section A.1) of Algorithm 1 on $N \times N$ activation matrices, for $N \in \{256 \cdot i \mid i \in \mathbb{Z}, 2 \leq i \leq 32\}$. Note that $N = 256 \cdot 32 = 8192$ corresponds to the dimension of LLaMA-65B (Touvron et al., 2023), an important LLM benchmark. Figure 4a shows that when $N = 8192$, our BBQ quantization kernel takes 0.3 milliseconds to quantize an $N \times N$ activation matrix to FP4. However, Figure 4b shows the latency saving of FP4 matrix multiplication over FP16 matrix multiplication is 4 milliseconds when $N = 8192$. Thus, for LLaMA-65B, BBQ's activation quantization adds a latency overhead that is only one-tenth the latency savings enabled by BBQ's reduced precision matrix multiplication. We attribute the low latency of the activation quantization kernel to the following: since quantization kernels are element-wise and highly memory-bound, compute units on GPUs are often idle waiting for data to arrive from global memory, and therefore additional computation (such as the Hadamard transform and binary search) can be fused into the quantization kernel without incurring latency overhead.

We next show BBQ can outperform FP16 and NF4 in terms of end-to-end inference latency, using LLaMA-700M and LLaMA-2.4B as our benchmarks. Figure 5 illustrates the overall inference latency, with contributions from different types of kernels shown in different colours. While BBQ adds

| Row | HT | RMS | PIT | Learn $\gamma$ | $\gamma$ Init | Perplexity | Entropy | Notes |
|-----|-----|-----|-----|---------|--------|------------|---------|-------|
| 1 | ✓ | ✓ | ✓ | ✓ | ✓ | 31.34 | 1.97 | BBQ |
| 2 | ✗ | ✓ | ✓ | ✓ | ✓ | 35.79 | 1.98 | |
| 3 | ✓ | ✗ | ✓ | ✓ | ✓ | 35.93 | 1.98 | |
| 4 | ✓ | ✓ | ✗ | ✗ | ✗ | 35.58 | 1.92 | QuEST |
| 5 | ✓ | ✓ | ✓ | ✗ | ✗ | 138.3 | 1.92 | |
| 6 | ✓ | ✓ | ✓ | ✗ | ✓ | 31.46 | 1.98 | |

Table 5: Ablation of BBQ features: Hadamard Transform (HT), RMS Normalization (RMS), Probability Integral Transform (PIT), making $\gamma$ a learnable parameter (Learn $\gamma$), and $\gamma$ initialization ($\gamma$ Init). A ✓ means a feature is present, and ✗ means the feature is not included.

additional latency overhead to quantize activations (green region) and perform element-wise scaling (orange region), BBQ significantly reduces the latency of matrix multiplication (blue region), which is the latency bottleneck in both FP16 and NF4. In general, BBQ shows a 40% speedup over FP16 and a 48% speedup over NF4. While both BBQ and NF4 can compress large models, BBQ is compute-efficient and has shorter latency than FP16. In contrast, during prefill, NF4 has longer latency than FP16, since NF4-quantized activations and weights must be de-quantized before being consumed by full-precision matrix multiplication kernels.

## 4.3 ABLATION

In this section, we present ablation studies of BBQ's individual features in Table 5. Specifically, we study the effects of Hadamard Transform, RMS Normalization, probability integral transform ($\Phi$) instead of the regular clipping function, having a learnable scaling factor $\gamma$, and initializing $\gamma$ to $\zeta^* \sigma_0$ instead of a dummy value (e.g. 1). We evaluate on a 2-bit LLaMA-95M pre-trained with 3B tokens. Results show Hadamard Transform and RMS normalization are two solid features BBQ inherited from QuEST, as removing them from BBQ results in a perplexity increase of 4.45 (row 2 compared to row 1) and 4.59 (row 3 compared to row 1). The probability integral transform ($\Phi$) is meant to replace QuEST's clip function. We see that naively substituting clip with $\Phi$ (row 4 compared to row 5) without $\gamma$ initialization leads to divergence (a perplexity increase from 35.58 to 138.3), but the combination of $\Phi$ and $\gamma$ initialization (row 6) can outperform QuEST (row 4) by a perplexity decrease of 4.12. Lastly, making $\gamma$ learnable (row 1 compared to row 6) can further reduce the perplexity by a smaller margin of 0.12.

## 5 LIMITATIONS AND FUTURE WORK

While BBQ can achieve lower perplexity than QuEST and LSQ for **QAPT**, BBQ is, by design, a quantizer that makes no attempt to reduce/bound the Euclidean distance between $x$ and $\hat{x}$, since they live in separate domains. Therefore, when applied to **QAFT**, BBQ's unbounded quantization error will drastically reduce model quality and cannot catch up to "same-domain" quantizers (ones that follow Equation 1) like QuEST and LSQ within the short time frame of QAFT. For similar reasons, BBQ is not suitable for **PTQ**.

BBQ can maximize entropy under the assumption that the Hadamard Transform can Gaussianize (Panferov et al., 2025; Yang et al., 2025) BBQ's input data. Empirically, we find $\mathrm{HT}(x)$ to be exactly Gaussian at the beginning of training, but not perfectly Gaussian afterwards. As future work, it may be possible to replace $\Phi$ with a more accurate smooth approximation of the non-differentiable empirical CDF of $\mathrm{HT}(x)$, to more closely approximate true probability integral transform.

## 6 CONCLUSION

In this work, we identify existing SOTA QAPT methods under-utilize learning capacity. While existing ITO quantization methods can maximize entropy, they are not compute-efficient, which limits their applicability to edge devices with energy constraints. Utilizing our key insight that learning is domain-agnostic, we propose BBQ, which performs ITO quantization in the input domain, while returning outputs in a compute-efficient domain. Empirical results show BBQ outperforms QuEST and LSQ without sacrificing compute efficiency.

## 7 REPRODUCIBILITY STATEMENT

A link to our source code is provided in the abstract of the paper. Our code should allow readers to reproduce tables and figures in the paper.

## 8 ACKNOWLEDGMENTS

This research was supported by the Natural Sciences and Engineering Research Council of Canada through a Discovery Grant. We thank Renjie Liao for providing access to additional GPU resources at UBC. Tor Aamodt has recently served as a consultant for NVIDIA, HPE, Cisco and IBM.

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

## A   APPENDIX

### A.1   KERNEL IMPLEMENTATION

In this section, we present an example implementation of Algorithm 1 using the Triton (Tillet et al., 2019) programming language.

Nvidia GPUs provide extensive support for low-precision tensor core operations, allowing expensive high-precision matrix multiplications to be replaced with low-precision alternatives, leading to higher compute throughput and lower energy consumption. For example, the Nvidia Blackwell architecture (Nvidia, 2025) provides tensor core support for the MX FP4(Rouhani et al., 2023) data type. As another example, the Turing (Nvidia, 2018) and Ampere (Nvidia, 2021) architectures provide support for the INT4 data type. We list in Table 6 the representable values for both data types.

| Binary Representation | Signed INT4 | MX FP4 |
|---|---|---|
| 0b0000 | 0 | 0 |
| 0b0001 | 1 | 0.5 |
| 0b0010 | 2 | 1 |
| 0b0011 | 3 | 1.5 |
| 0b0100 | 4 | 2 |
| 0b0101 | 5 | 3 |
| 0b0110 | 6 | 4 |
| 0b0111 | 7 | 6 |
| 0b1000 | -8 | 0 |
| 0b1001 | -7 | -0.5 |
| 0b1010 | -6 | -1 |
| 0b1011 | -5 | -1.5 |
| 0b1100 | -4 | -2 |
| 0b1101 | -3 | -3 |
| 0b1110 | -2 | -4 |
| 0b1111 | -1 | -6 |

Table 6: Signed INT4 vs. MX FP4

Table 7 shows the values that BBQ can represent.

| Precision | Possible Values of $q$ |
|---|---|
| 4 | -8,-7,-6,-5,-4,-3,-2,-1,0,1,2,3,4,5,6,7 |
| 3 | -4,-3,-2,-1,0,1,2,3 |
| 2 | -1.5,-0.5,0.5,1.5 |
| 1 | -0.5,0.5 |

Table 7: Values that BBQ can represent. As shown, for $b = 4$, BBQ can be encoded as INT4. For $b = 3$, BBQ can be encoded as INT4 or FP4. For $b \in \{1, 2\}$, BBQ can be encoded as FP4.

Since $b = 3$ can be encoded by both INT4 and FP4, we present an example triton kernel for BBQ with 3-bit quantization following Algorithm 1.

```
@triton.jit
def bbq(
    x_ptr,
    h_ptr,
    rsigma_ptr,
    qmz_ptr,
    ROW_SIZE: tl.constexpr,
    BLOCKSIZE: tl.constexpr=128,
    DTYPE: tl.constexpr = "int4"
):
    rid = tl.program_id(axis=0)
    cid = tl.program_id(axis=1)
    xrids = rid * BLOCKSIZE + tl.arange(0, BLOCKSIZE)[:, None]
    xcids = cid * BLOCKSIZE + tl.arange(0, BLOCKSIZE)[None, :]
    xids = xrids * ROW_SIZE + xcids

    hrids = tl.arange(0, BLOCKSIZE)[:, None]
    hcids = tl.arange(0, BLOCKSIZE)[None, :]
    hids = hrids * BLOCKSIZE + hcids

    # load a block of x
    x = tl.load(x_ptr + xids)
    # load pre-computed hadamard matrix shared across the entire kernel
    h = tl.load(h_ptr + hids)
    # load reciprocal of sigma
    rsigma = tl.load(rsigma_ptr)

    # hadamard transform
    v = tl.dot(x, h) * rsigma

    # accelerated normal CDF and rounding
    qmz = normal_cdf_and_round_3bit(v, DTYPE)

    # pack two 4-bit data type into a single byte
    shift = tl.arange(0, 2)[None, None, :] * 4
    qmz = tl.reshape(qmz, (BLOCKSIZE, BLOCKSIZE // 2, 2)) << shift
    qmz = tl.xor_sum(qmz, axis=-1, keep_dims=False)

    # write quantized data to memory
    qmzrids = rid * BLOCKSIZE + tl.arange(0, BLOCKSIZE)[:, None]
    qmzcids = cid * (BLOCKSIZE // 2) + tl.arange(0, BLOCKSIZE // 2)[None, :]
    qmzids = qmzrids * (ROW_SIZE // 2) + qmzcids
    tl.store(qmz_ptr + qmzids, qmz)
```

Next, we show the definition of function normal_cdf_and_round_3bit. We pre-compute $\Phi^{-1}\left(\frac{i}{2^b}\right)$ for all possible values of $i$, and use a binary search method to decide which $i$ should be the result of quantization, and lastly, directly use the binary representation of $i - 2^{b-1} - z$ according to Table 6. For 3-bit quantization, the 8 pre-computed values of $\Phi^{-1}\left(\frac{i}{2^b}\right)$, for $i \in \{0, 1, 2, 3, 4, 5, 6, 7\}$, are {-∞, -1.1503493803760083, -0.6744897501960818, -0.3186393639643752, 0.0, 0.3186393639643752, 0.6744897501960818, 1.1503493803760083}. The corresponding values of $i - 2^{b-1} - z$ are $\{-4, -3, -2, -1, 0, 1, 2, 3\}$.

```python
@triton.jit
def normal_cdf_and_round_3bit(v: tl.tensor, DTYPE: tl.constexpr):
    if DTYPE == "int4":
        qmz = tl.where(
            v >= 0.0,
            tl.where(
                v >= 0.6744897501960818,
                tl.where(v >= 1.1503493803760083, 0b0011, 0b0010),
                tl.where(v >= 0.3186393639643752, 0b0001, 0b0000),
            ),
            tl.where(
                v >= -0.6744897501960818,
                tl.where(v >= -0.3186393639643752, 0b1111, 0b1110),
                tl.where(v >= -1.1503493803760083, 0b1101, 0b1100),
            )
        ).to(tl.int8)
        return qmz
    elif DTYPE == "fp4":
        qmz = tl.where(
            v >= 0.0,
            tl.where(
                v >= 0.6744897501960818,
                tl.where(v >= 1.1503493803760083, 0b0101, 0b0100),
                tl.where(v >= 0.3186393639643752, 0b0010, 0b0000),
            ),
            tl.where(
                v >= -0.6744897501960818,
                tl.where(v >= -0.3186393639643752, 0b1010, 0b1100),
                tl.where(v >= -1.1503493803760083, 0b1101, 0b1110),
            )
        ).to(tl.int8)
        return qmz
```

## A.2 EXPLANATION OF EQUATION 9

In this section, we discuss why Equation 9 is useful in ensuring that elements of $x$ and elements of $\hat{x}$ have approximately the same average magnitude. We first re-iterate the definitions of $\hat{x}$:

$$
\begin{aligned}
v &= \text{HT}(x)/\sigma \\
q &= \lfloor 2^b \Phi(v) \rfloor - 2^{b-1} - z \\
\hat{x} &= \frac{\gamma}{2^{b-1}} q
\end{aligned}
\tag{10}
$$

We first assume elements of $\text{HT}(x)$ has approximately the same average magnitude as elements of $x$, since the Hadamard matrix is orthogonal. This means elements of $v = \text{HT}(x)/\sigma$ has an average magnitude that is $\sigma$ times smaller than that of $x$. We next remove the floor operation and the zero point $z$ for simplicity, arriving at the following simplified equation:

$$
\begin{aligned}
v &= \text{HT}(x)/\sigma \\
q &= 2^b \Phi(v) - 2^{b-1} \\
\hat{x} &= \frac{\gamma}{2^{b-1}} q = \gamma(2\Phi(v) - 1)
\end{aligned}
\tag{11}
$$

We assume the average magnitude of elements of $2\Phi(v) - 1$ to be $\zeta^*$ times smaller than that of $v$. In other words, the average magnitude of elements of $2\Phi(v) - 1$ is approximately $\zeta^*\sigma$ times smaller than that of $x$. Therefore, we compensate by setting $\gamma = \zeta^*\sigma$, ensuring elements of $\hat{x} = \gamma(2\Phi(v) - 1)$ has approximately the same average magnitude as $x$.

To estimate the value of $\zeta^*$, we use Equation 9, which expresses $\zeta^*$ as the optimal $\zeta$ that minimizes the MSE between $v$ and $\zeta(2\Phi(v) - 1)$ when $v$ follows a Gaussian distribution. Using Monte-Carlo sampling to estimate the MSE, and gradient descent to solve for the $\arg\min$, we estimate $\zeta^* \approx 1.694$.

## A.3 ABLATION ON EXPONENTIAL MOVING AVERAGE OF $1/\sigma$

In this section, we show BBQ and BBQ-Fast achieve identical perplexity.

|         | 4-bit | 3-bit | 2-bit | 1-bit |
|---------|-------|-------|-------|-------|
| BBQ     | 25.51 | 26.55 | 31.35 | 49.80 |
| BBQ-Fast | 25.40 | 26.55 | 31.43 | 49.72 |

Table 8: Perplexity of BBQ vs BBQ-Fast, on LLaMA-95M pre-trained with 3 billion C4 tokens.

## A.4 Zero-shot Results

In this section, we present the zero-shot evaluation perplexity in Table 9, which corresponds to the models pre-trained in Table 3.

| Model | Dataset | Params n+e | Method | Bits WA | Perplexity |
|-------|---------|------------|--------|---------|------------|
| LLaMA | wikitext | 95M | None | 16 | 56.11 |
| LLaMA | wikitext | 95M | BBQ | 4 | **57.87** |
| LLaMA | wikitext | 95M | QuEST | 4 | 58.91 |
| LLaMA | wikitext | 95M | LSQ | 4 | 61.62 |
| LLaMA | wikitext | 95M | BBQ | 3 | **60.18** |
| LLaMA | wikitext | 95M | QuEST | 3 | 64.21 |
| LLaMA | wikitext | 95M | LSQ | 3 | 68.35 |
| LLaMA | wikitext | 95M | BBQ | 2 | **70.19** |
| LLaMA | wikitext | 95M | QuEST | 2 | 77.32 |
| LLaMA | wikitext | 95M | LSQ | 2 | 80.11 |
| LLaMA | wikitext | 95M | BBQ | 1 | **109.66** |
| LLaMA | wikitext | 95M | QuEST | 1 | 172.77 |
| LLaMA | wikitext | 125M | None | 16 | 50.26 |
| LLaMA | wikitext | 125M | BBQ | 4 | **50.50** |
| LLaMA | wikitext | 125M | QuEST | 4 | 51.31 |
| LLaMA | wikitext | 125M | LSQ | 4 | 53.72 |
| LLaMA | wikitext | 125M | BBQ | 3 | **50.94** |
| LLaMA | wikitext | 125M | QuEST | 3 | 54.37 |
| LLaMA | wikitext | 125M | LSQ | 3 | 58.61 |
| LLaMA | wikitext | 125M | BBQ | 2 | **60.47** |
| LLaMA | wikitext | 125M | QuEST | 2 | 69.45 |
| LLaMA | wikitext | 125M | LSQ | 2 | 70.61 |
| LLaMA | wikitext | 125M | BBQ | 1 | **97.34** |
| LLaMA | wikitext | 125M | QuEST | 1 | 180.33 |
| LLaMA | wikitext | 200M | None | 16 | 40.42 |
| LLaMA | wikitext | 200M | BBQ | 4 | 42.56 |
| LLaMA | wikitext | 200M | QuEST | 4 | **41.67** |
| LLaMA | wikitext | 200M | LSQ | 4 | 4133 |
| LLaMA | wikitext | 200M | BBQ | 3 | **41.82** |
| LLaMA | wikitext | 200M | QuEST | 3 | 43.95 |
| LLaMA | wikitext | 200M | LSQ | 3 | 426.8 |
| LLaMA | wikitext | 200M | BBQ | 2 | **49.53** |
| LLaMA | wikitext | 200M | QuEST | 2 | 54.12 |
| LLaMA | wikitext | 200M | LSQ | 2 | 208.5 |
| LLaMA | wikitext | 200M | BBQ | 1 | **80.34** |
| LLaMA | wikitext | 200M | QuEST | 1 | 123.19 |
| LLaMA | wikitext | 300M | None | 16 | 34.95 |
| LLaMA | wikitext | 300M | BBQ | 4 | 38.57 |
| LLaMA | wikitext | 300M | QuEST | 4 | **35.26** |
| LLaMA | wikitext | 300M | BBQ | 3 | **36.77** |
| LLaMA | wikitext | 300M | QuEST | 3 | 37.26 |
| LLaMA | wikitext | 300M | BBQ | 2 | **41.24** |
| LLaMA | wikitext | 300M | QuEST | 2 | 45.28 |

Table 9: Zero-shot wikitext perplexity.

A.5 BBQ VS. QUEST

In this section, we illustrate the difference between BBQ and QuEST in Figure 6.

- BBQ and QuEST share the Hadamard transform (①) and RMS normalization (②) to re-shape the distribution to better match the standard Gaussian.

- In step ③, QuEST scales by $\alpha^*$, shifts by 0.5, and uses the clip function to limit values within a finite range. BBQ uses the $\Phi$ function to limit values within a finite range, while simultaneously maximizing the entropy.

- In step ④, both methods use uniform quantization to quantize data. Both methods ensure the quantized data can be stored in a compute-efficient data type (integer).

- In step ⑤, BBQ applies linear scaling, while QuEST reverses all of the operations it did before (RMS Normalization, scaling by $\alpha^*$, and shifting by 0.5), as QuEST is a "same-domain" quantizer. BBQ is a "cross-domain" quantizer, so BBQ does not reverse its previous operation such as $\Phi$, shifting by $2^b - 1$ and $z$, multiplication by $2^b$, division by $\sigma$ etc. Instead, BBQ applies linear scaling to scale its range to be within $[-\gamma, \gamma]$ where $\gamma$ is a learnable parameter.

- In step ⑥, QuEST reverses its Hadamard transform (a unitary operation). BBQ does not reverse the Hadamard transform because it is a "cross-domain" quantizer.

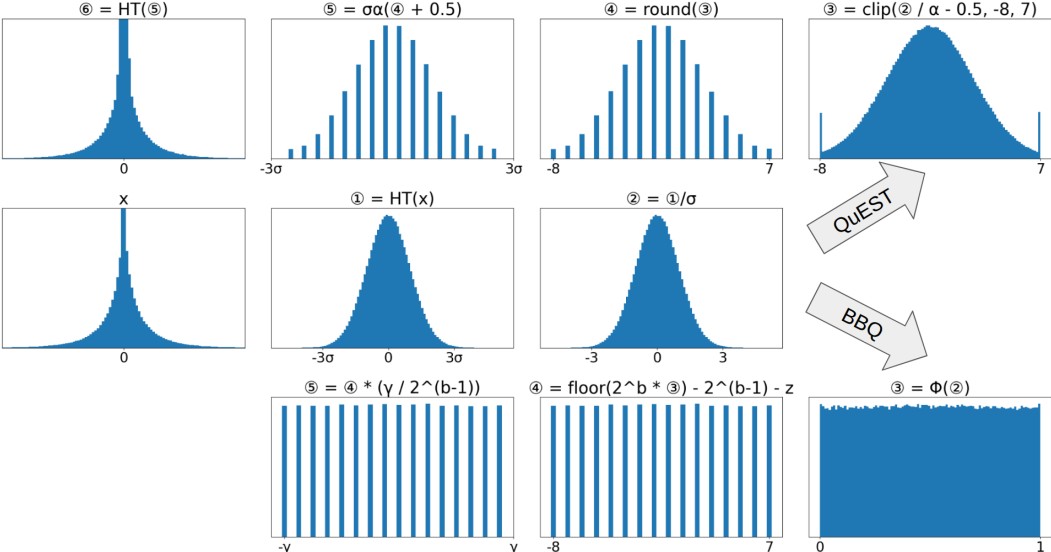

Figure 6: BBQ vs. QuEST.

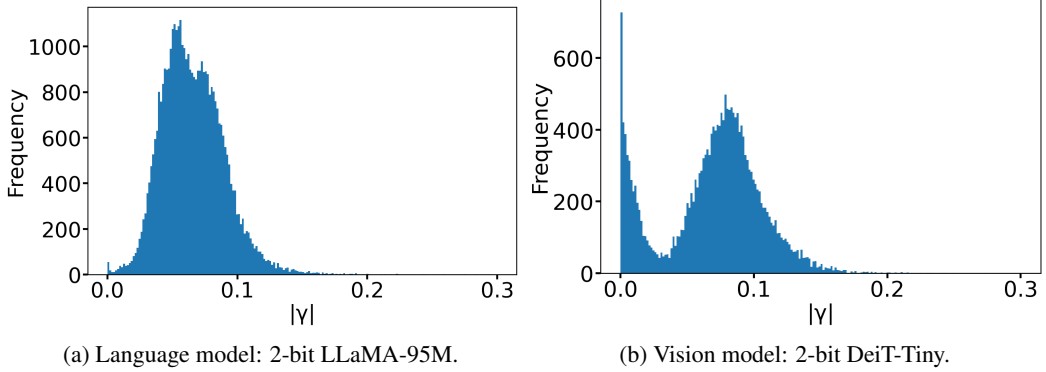

(a) Language model: 2-bit LLaMA-95M.  (b) Vision model: 2-bit DeiT-Tiny.

Figure 7: Histogram of the absolute value of learned $\gamma$ in a language model versus a vision model.

## A.6 EXTENSION TO VISION MODELS

While BBQ is designed for language models, we discuss and evaluate a potential extension of BBQ to vision models. When we naively apply BBQ to vision models without any modification, we notice, as shown in Figure 7b, a subset of learned $\gamma$ have magnitudes extremely close to 0. Since every $\gamma$ is assigned to a weight channel, if $\gamma$ is small, weights from the corresponding channel suffers from gradient vanishing. As shown in Figure 7a, language models do not suffer from such $\gamma$ collapsing. Instead of making $\gamma$ a learnable parameter and initializing $\gamma$ to $\zeta^* \sigma_0$ as discussed in Section 3.4, we propose a BBQ variant, BBQ-Vision, that instead dynamically calculates the value of $\gamma$ as $\zeta \sigma$ on every forward pass, where $\zeta$ is a constant hyperparameter. BBQ-Vision prevents $\gamma$ from collapsing to 0, since for $\gamma$ to be 0, all weights in that channel must also be 0. In addition, BBQ-Vision still preserves some degree of learnability as the network can modify weights/activations to indirectly control the value of $\sigma$. We compare BBQ-Vision against QuEST and LSQ on DeiT (Touvron et al., 2021) and Resnet (He et al., 2015) with various sizes and show our results in Table 10. We use $\zeta = 2.45$. Our preliminary results show BBQ-Vision can consistently outperform QuEST and LSQ, especially at lower precisions (1-bit and 2-bit).

| Model | Params | Bits WA | BBQ-Vision | | | QuEST | | | LSQ | | |
|---|---|---|---|---|---|---|---|---|---|---|---|
| | | | Ent | TL | VA | Ent | TL | VA | Ent | TL | VA |
| DEIT-T | 5M | 4 | **3.85** | **2.84** | 74.68 | 3.61 | 2.91 | 74.3 | 3.67 | 2.88 | **74.88** |
| DEIT-T | 5M | 3 | **2.93** | **2.89** | **74.24** | 2.78 | 2.98 | 72.2 | 2.81 | 3.01 | 71.36 |
| DEIT-T | 5M | 2 | **1.94** | **3.05** | **70.16** | 1.93 | 3.18 | 67.24 | 1.90 | 3.23 | 65.32 |
| DEIT-T | 5M | 1 | **1.00** | **3.52** | **53.54** | **1.00** | 3.67 | 47.55 | - | - | - |
| DEIT-S | 20M | 4 | **3.82** | 2.53 | 79.46 | 3.61 | **2.51** | 80.00 | 3.57 | 2.52 | **80.04** |
| DEIT-S | 20M | 3 | **2.91** | **2.48** | **80.88** | 2.78 | 2.62 | 79.34 | 2.74 | 2.64 | 78.96 |
| DEIT-S | 20M | 2 | **1.93** | **2.76** | **77.02** | 1.93 | 2.82 | 76.18 | 1.83 | 2.87 | 74.42 |
| DEIT-S | 20M | 1 | **1.00** | **3.18** | **66.16** | 1.00 | 3.37 | 59.94 | - | - | - |
| Resnet-10 | 5M | 1 | **1.00** | **3.37** | **63.6** | 1.00 | 3.40 | 62.4 | - | - | - |
| Resnet-18 | 11M | 1 | **1.00** | **3.12** | **73.13** | 1.00 | 3.14 | 72.28 | - | - | - |

Table 10: Entropy (Ent), Training Cross Entropy Loss (TL), and Validation top-1 Accuracy (VA) of vision models pretrained on Imagenet-100 (Chun-Hsiao Yeh, 2022; Russakovsky et al., 2015).

## A.7 VALIDATION LOSS VS. TRAINING PROGRESS

In this section we present validation loss vs. training iteration curves for all of our experiments in Section 4. For visualization purposes, we only show the validation loss for the last 90% iterations, hide the first 10% iterations when learning rate is warming up, and don't show loss curves for methods that diverged. For all figures, the y-axis is the validation cross entropy loss, and x-axis is iterations (batches). For all experiments, we quantize both weights and activations of linear layers to $b$ bits. In addition, for all QuEST experiments, we use a trust factor of $T = \alpha^*/(2^b - 1)$ as discuss in QuEST (Panferov et al., 2025).

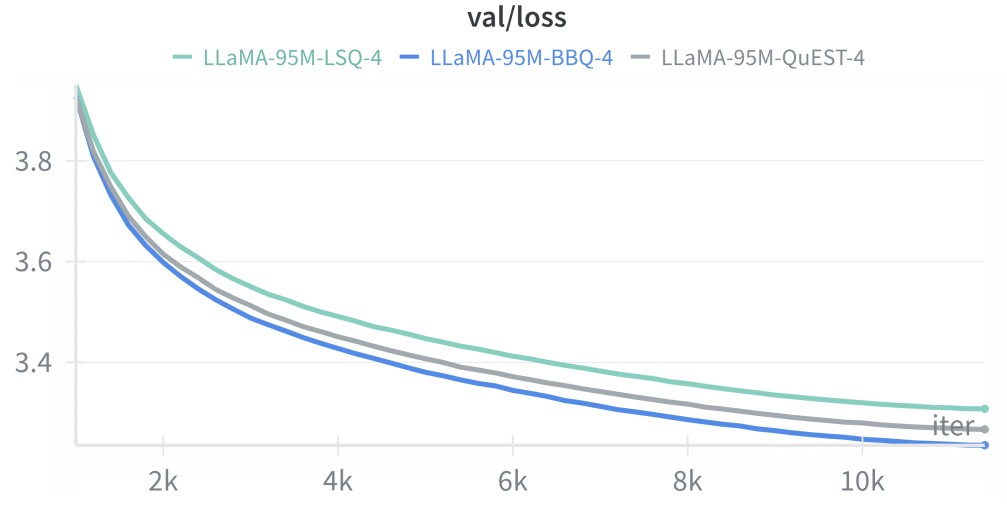

Figure 8: LLaMA-95M (4-bit) pre-trained on 3 billion C4 tokens (batched over 12 thousand iterations). LSQ is green, QuEST is gray, and BBQ is blue.

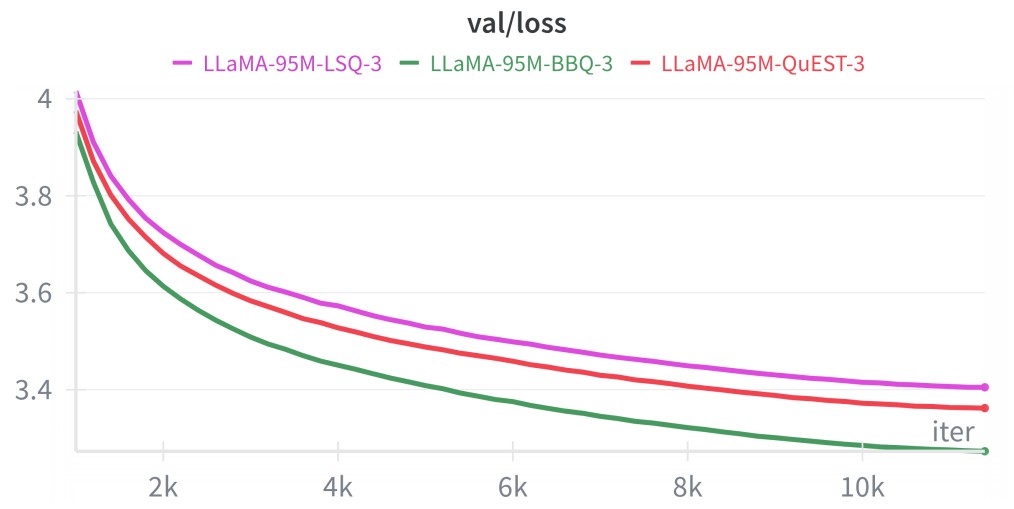

Figure 9: LLaMA-95M (3-bit) pre-trained on 3 billion C4 tokens (batched over 12 thousand iterations). LSQ is pink, QuEST is red, and BBQ is green.

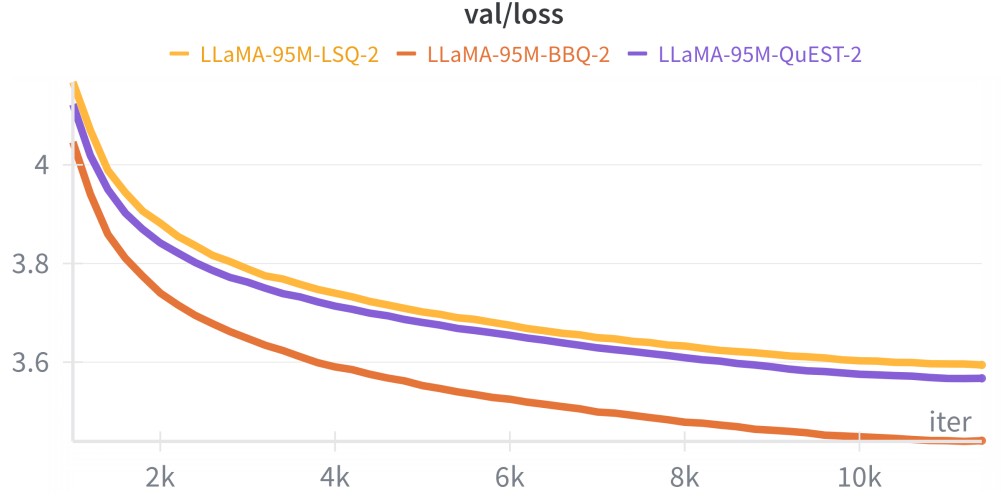

Figure 10: LLaMA-95M (2-bit) pre-trained on 3 billion C4 tokens (batched over 12 thousand iterations). LSQ is yellow, QuEST is purple, and BBQ is orange.

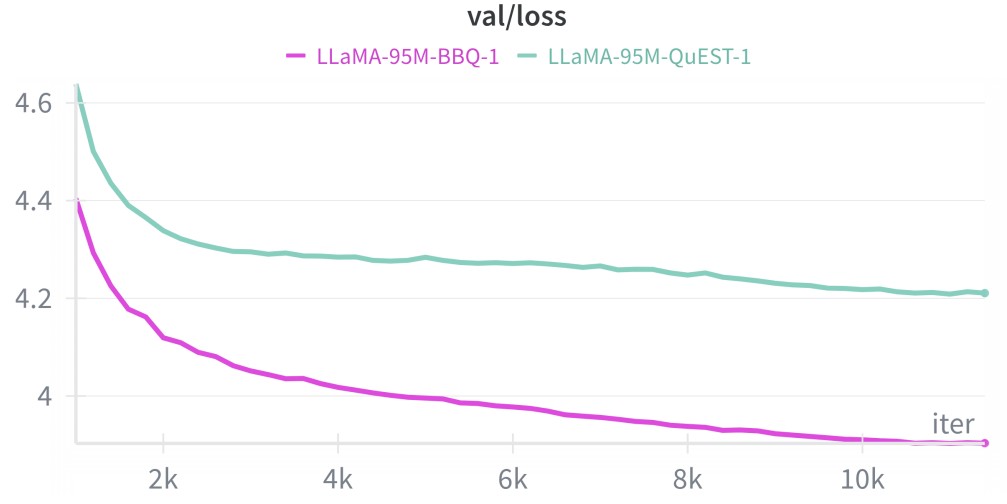

Figure 11: LLaMA-95M (1-bit) pre-trained on 3 billion C4 tokens (batched over 12 thousand iterations). QuEST is purple and BBQ is orange.

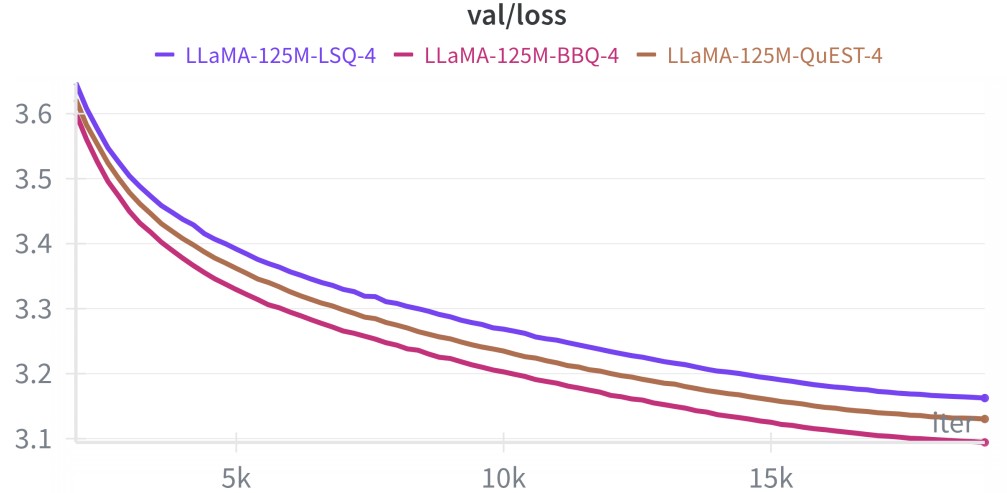

Figure 12: LLaMA-125M (4-bit) pre-trained on 5 billion C4 tokens (batched over 20 thousand iterations). LSQ is red, QuEST is brown, and BBQ is pink.

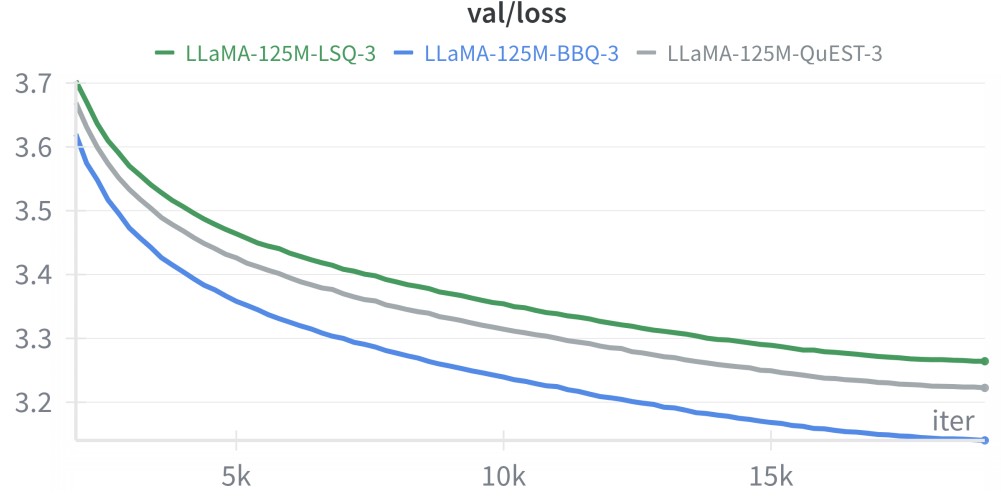

Figure 13: LLaMA-125M (3-bit) pre-trained on 5 billion C4 tokens (batched over 20 thousand iterations). LSQ is green, QuEST is gray, and BBQ is blue.

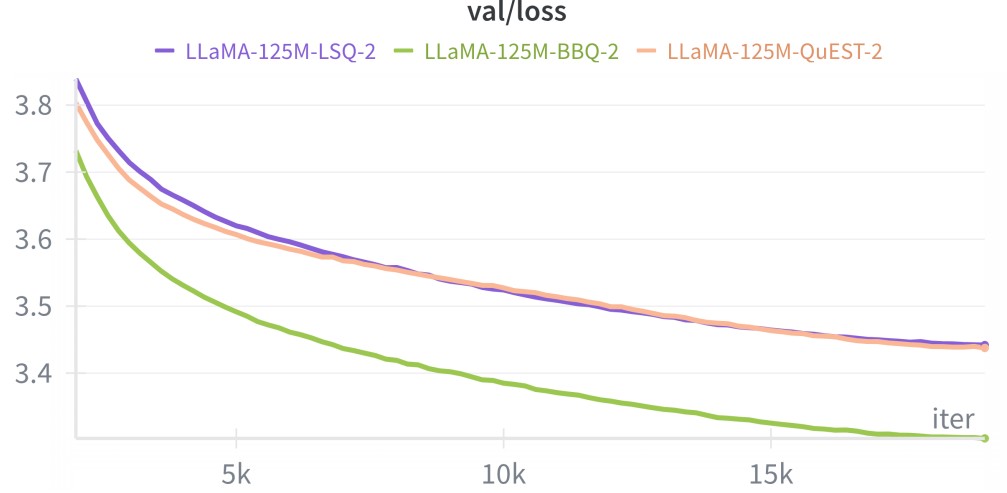

Figure 14: LLaMA-125M (2-bit) pre-trained on 5 billion C4 tokens (batched over 20 thousand iterations). LSQ is purple, QuEST is orange, and BBQ is green.

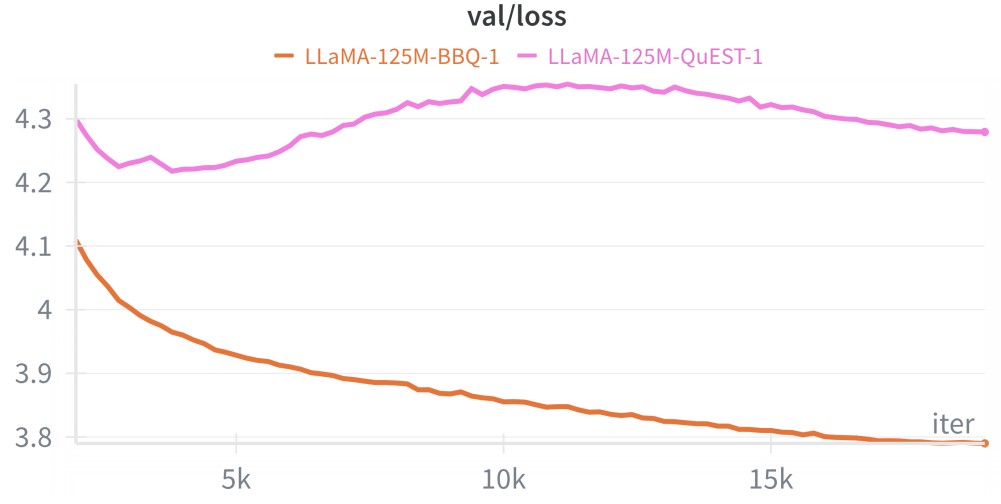

Figure 15: LLaMA-125M (1-bit) pre-trained on 5 billion C4 tokens (batched over 20 thousand iterations). QuEST is pink and BBQ is orange.

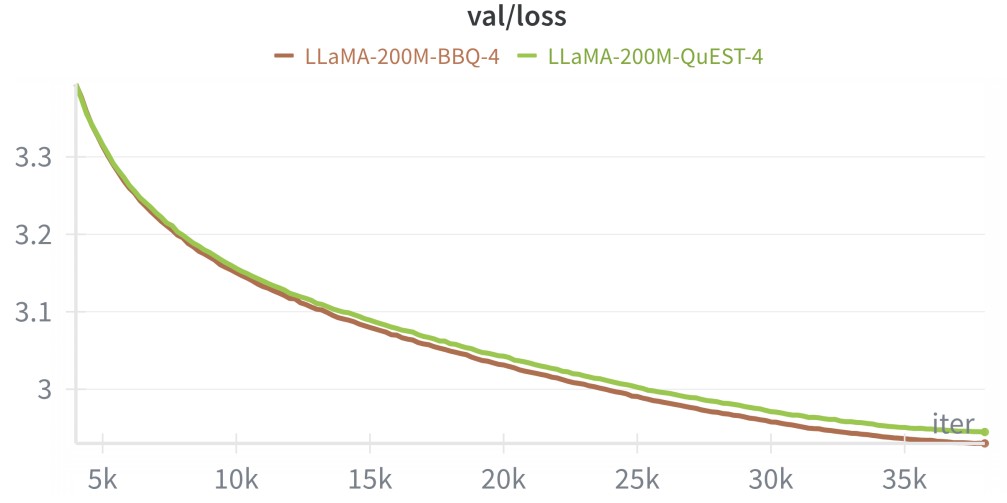

Figure 16: LLaMA-200M (4-bit) pre-trained on 10 billion C4 tokens (batched over 40 thousand iterations). QuEST is green and BBQ is brown.

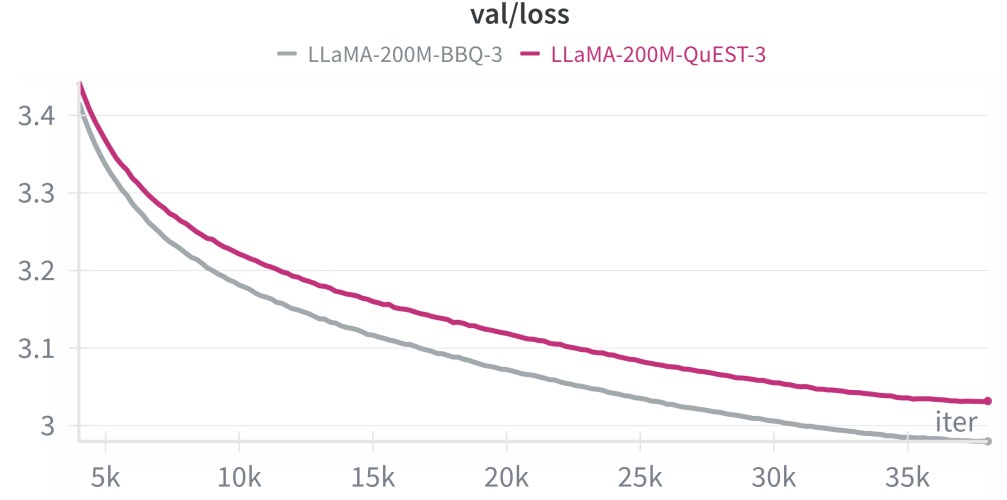

Figure 17: LLaMA-200M (3-bit) pre-trained on 10 billion C4 tokens (batched over 40 thousand iterations). QuEST is pink and BBQ is gray.

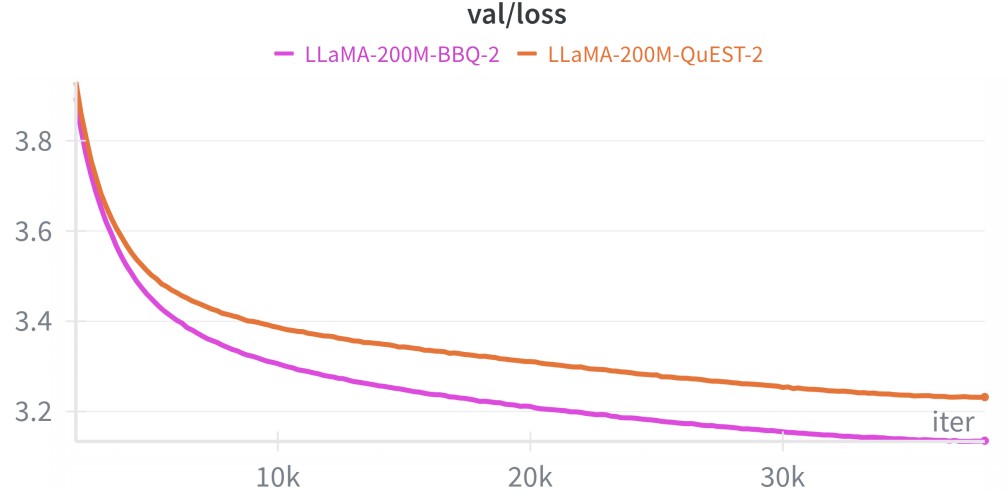

Figure 18: LLaMA-200M (2-bit) pre-trained on 10 billion C4 tokens (batched over 40 thousand iterations). QuEST is orange and BBQ is pink.

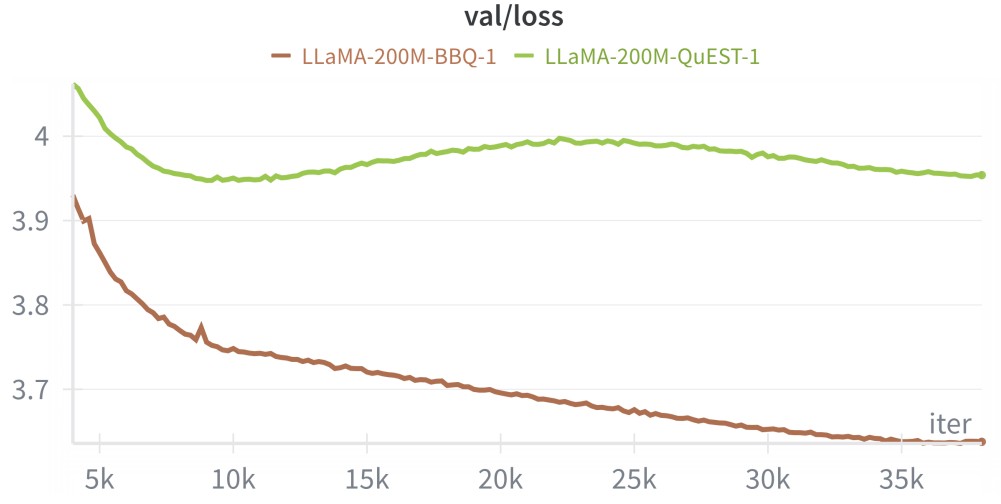

Figure 19: LLaMA-200M (1-bit) pre-trained on 10 billion C4 tokens (batched over 40 thousand iterations). QuEST is green and BBQ is brown.

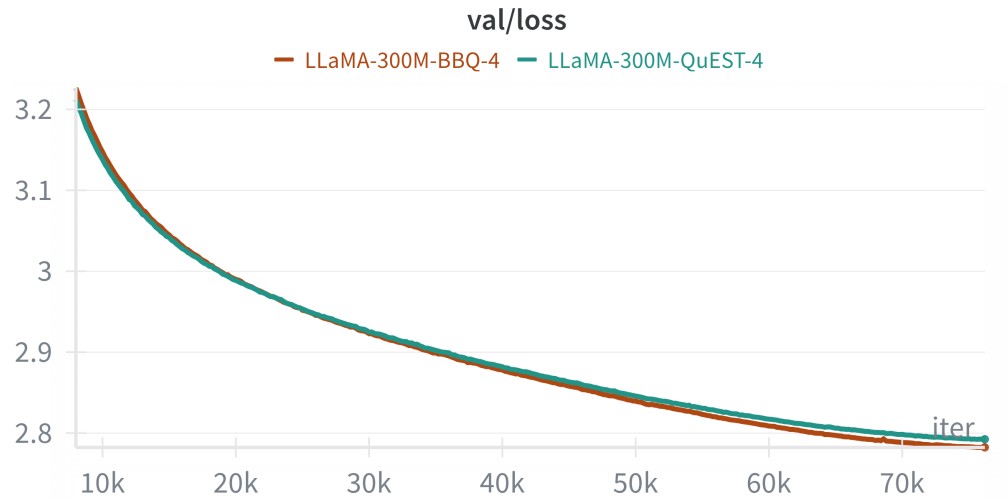

Figure 20: LLaMA-300M (4-bit) pre-trained on 20 billion C4 tokens (batched over 80 thousand iterations). QuEST is green and BBQ is brown.

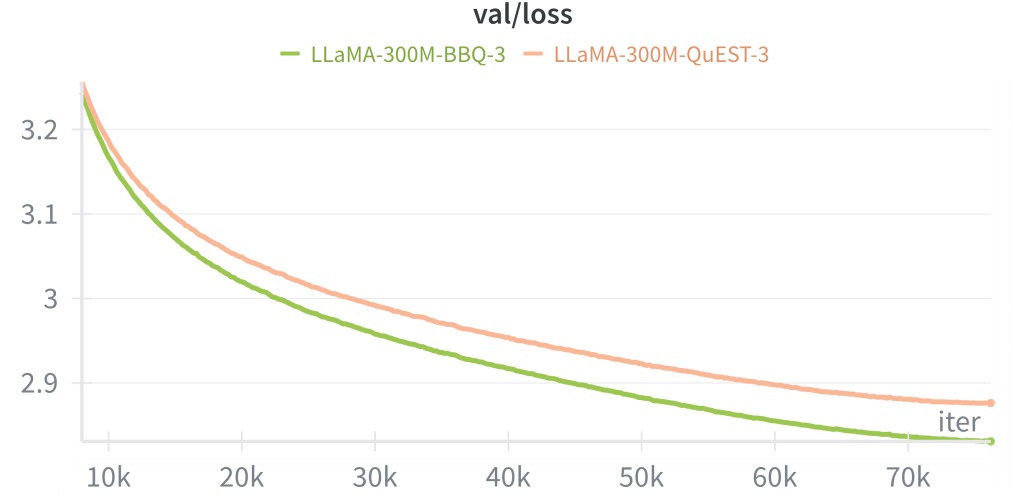

Figure 21: LLaMA-300M (3-bit) pre-trained on 20 billion C4 tokens (batched over 80 thousand iterations). QuEST is orange and BBQ is green.

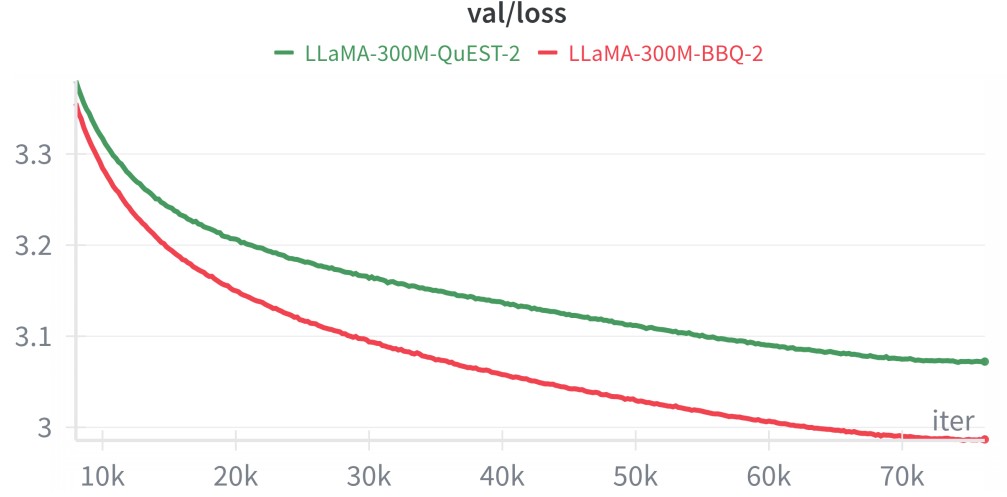

Figure 22: LLaMA-300M (2-bit) pre-trained on 20 billion C4 tokens (batched over 80 thousand iterations). QuEST is green and BBQ is red.

