# OpenReview forum: "Boosting Entropy with Bell Box Quantization"
_ICLR.cc/2026/Conference — ICLR 2026 Poster_

### Official Review · Reviewer_VA2y · 2025-10-29

**Soundness:** 2
**Presentation:** 2
**Contribution:** 2
**Rating:** 4
**Confidence:** 4

**Summary:**

This paper proposes BBQ, a novel quantization method for low-bit training of large language models. The authors improve the quantization process via adaptive scaling factors and demonstrate its performance on the LLaMA model under 1-4 bit quantization, comparing it with QuEST/LSQ methods.

**Strengths:**

(1) Propose the BBQ quantization method, which optimizes low-bit quantization through adaptive scaling factors
(2) Systematically evaluate quantization performance across different bit widths (1-4 bit) on the LLaMA model

**Weaknesses:**

(1) Limited experimental scale and diversity: The method is tested only on LLaMA, without validation on other mainstream architectures (e.g., GPT, Bloom), making it hard to confirm its generalizability
(2) Insufficient theoretical analysis: No in-depth explanation is provided for why BBQ performs better at certain bit widths (e.g., 3-bit), relying solely on experimental observations without theoretical support

**Questions:**

(1) The authors mention that "LSQ does not support 1-bit quantization and LSQ diverges when n+ e= 300M". Does BBQ also face similar limitations? Will BBQ remain effective for models with larger parameter scales (e.g., >300M)?
(2) The paper states that "each experiment for n+ e= 300M is conducted on one Nvidia A100 80GB and lasts for 3.5 days". For larger-scale models (e.g., 1B+ parameters), will the computational cost of BBQ increase significantly? Is it still practically viable?

---

> ### Author Response · Authors · 2025-11-22
>
> We thank Reviewer VA2y for reviewing our paper and for providing feedback. We are glad you find our evaluation systematic.
>
> ### 1. Expanding the Experimentation section
>
> > Limited experimental scale and diversity: The method is tested only on LLaMA, without validation on other mainstream architectures (e.g., GPT, Bloom), making it hard to confirm its generalizability
>
> **Response** Following your suggestion, we have extensively expanded the experimentation section to additionally include results of models with the GPT architecture (in Table 4), instead of just having the models with the LLaMA architecture. Results (Table 3 and 4) show BBQ consistently outperforms other SOTA methods on both LLaMA and GPT.
>
> **Action 1.1** We added QAPT results on the GPT architecture in Table 4, in addition to the LLaMA results in Table 3..
>
> ### 2. In-depth explanation
>
> > Insufficient theoretical analysis: No in-depth explanation is provided for why BBQ performs better at certain bit widths (e.g., 3-bit), relying solely on experimental observations without theoretical support
>
> BBQ is an empirical study. We do not propose new theorems that require theoretical analysis. In the field of Machine Learning, many influential papers are empirical studies. For example,  “Attention is all you need” and “Deep Residual Learning for Image Recognition” are two influential papers that proposed new model architectures and showed empirical evidence that the proposed methods work well without deep theoretical insights and mathematical proofs.
>
> BBQ is built on the intuitive insight that if we increase the entropy of weights/activations, we can alleviate the information bottleneck of models with limited memory footprint and help them accumulate more information from their training set, leading to improved accuracy. Multiple DNN quantization works [1][2] have confirmed that entropy positively correlates with model accuracy. Our experimental results in Table 3 serves as an empirical evidence to our intuitive insight: BBQ achieves higher entropy than all prior methods and lower perplexity than prior methods.
>
> [1] Ecco: Improving memory bandwidth and capacity for llms via entropy-aware cache compression.
>
> [2] Edgeqat: Entropy and distribution guided quantization-aware training for the acceleration of lightweight llms on the edge.
>
>
> ### 3. Clarification
>
> > The paper states that "each experiment for n+ e= 300M is conducted on one Nvidia A100 80GB and lasts for 3.5 days". For larger-scale models (e.g., 1B+ parameters), will the computational cost of BBQ increase significantly? Is it still practically viable?
>
> We note that BBQ is a quantization-aware pre-training algorithm, and pre-training is known to be expensive and not affordable in non-corporate settings, especially for large models. Compared to other pre-training algorithms such as QuEST, BBQ has similar pre-training computational cost. While for larger-scale models, the pre-training computational cost of BBQ will increase significantly, so will other pre-training algorithms. Therefore, BBQ is practically viable in the pre-training regime. Lastly, we note that our profiling results in Figure 6 shows BBQ can reduce the **inference** latency by 40% compared to baseline.
>
> > The authors mention that "LSQ does not support 1-bit quantization and LSQ diverges when n+ e= 300M". Does BBQ also face similar limitations? Will BBQ remain effective for models with larger parameter scales (e.g., >300M)?
>
> The answer to the first question is in Table 3 in the original submission. Table 3 shows BBQ works well with multiple 1-bit LLaMA models. Table 3 also shows BBQ works well with LLaMA-300M. Therefore, BBQ does not face the limitations of LSQ.
>
> While we believe BBQ will likely remain effective for larger parameter scales (> 300m), due to resource limitations, we cannot perform pre-training experiments at this scale. As discussed in the paper, the experiments in Table 3 already took us 1.5 GPU months to run and we are unable to afford more resource usage.

---

### Official Review · Reviewer_BCbT · 2025-10-29

**Soundness:** 3
**Presentation:** 3
**Contribution:** 2
**Rating:** 6
**Confidence:** 4

**Summary:**

This paper presents Bell Box Quantization (BBQ), a novel quantization method that aims to achieve information-theoretically optimal (ITO) quantization while remaining compute-efficient for modern hardware. The method introduces a seven-step quantization pipeline involving the Hadamard Transform, RMS Normalization, Gaussian CDF-based Probability Integral Transform, and precision-scaled quantization operations. BBQ is applied in Quantization-Aware Pre-Training (QAPT) settings, targeting low-precision training (1–4 bits) for large-scale models such as LLaMA. Experimental results show that BBQ outperforms existing quantization schemes like LSQ and QuEST in both entropy (capacity utilization) and perplexity (model quality), while maintaining hardware efficiency. The paper further demonstrates that entropy serves as a strong proxy for quantized model quality.

The paper presents a technically sound and conceptually novel quantization framework that advances the state of low-bit quantization for large-scale models. However, its current limitations in adaptability and architecture diversity need to be addressed in future work.

Recommendation: weak accept (with added generalization experiments and ablation analysis).

**Strengths:**

* The introduction of ITO-based quantization combined with hardware-compatible output domains is a well-motivated and original contribution to quantized training research.
* The step-by-step quantization process (Hadamard Transform → Gaussian CDF → Uniform Quantization → Scaling) is well explained and systematically justified.
* Across multiple model sizes and bit-widths, BBQ consistently achieves lower perplexity and higher entropy compared to LSQ and QuEST, validating the proposed method.
* Using entropy as a measurable proxy for model capacity utilization is a meaningful contribution that provides interpretability beyond standard performance metrics.
* The profiling experiments demonstrate that BBQ can achieve real-world speedups on hardware supporting FP4, showing its deployment potential for large models.

**Weaknesses:**

- section 2.3, "learning is domain agnostic" : the two example to support this is very simplistic assumption. For eg. rotated, cropped and color jitter image is still image and in the same domain, images transformed to frequency domain is still an image but encoded in different format. In both cases, domain is still image processing. Autoencoder is a bit closer but they can't be used to train cross domain models. Authors are suggested to add more concrete examples and possibly quantification of this assumption
Limited scope of evaluation: Experiments are conducted exclusively on LLaMA-based architectures; it remains unclear whether BBQ generalizes to convolutional or vision transformer models.
* Since BBQ operates in separate input/output domains and does not minimize reconstruction error, it may not perform well in fine-tuning or post-training quantization scenarios — limiting its versatility.
* Lack of ablation analysis: The impact of individual steps (e.g., Hadamard Transform, Φ smoothness, or γ initialization) is not quantitatively analyzed, leaving unclear which components are most critical.
* Implementation details underexplored: Although latency improvements are discussed, comparisons with more hardware-optimized kernels/different GPU generations  or newer architectures like NPUs could provide stronger evidence of scalability.
* No comparison to information-theoretic quantizers beyond LSQ/QuEST: Including recent ITO-based methods (e.g., [NF4](https://chatgpt.com/c/690279a0-9b50-8320-a3bc-93decef608ce#:~:text=neural%20network%20weights.-,arXiv,%2B2,-FP4%20(4%2Dbit), [FP4](https://arxiv.org/html/2501.17116v1?utm_source=chatgpt.com), or adaptive quantizers) would strengthen the empirical claim of superiority.

**Questions:**

- section 3.4 : Uniform quantization and unsigned to signed conversion : Does it make any difference for ReLU based models where output activations are always positive and unsigned values are preferred for better accuracy. ? I support this sentence "we subsequently convert the positive on data to symmetric data by subtracting 2b−1 and the hyperparameter zero point z," is to take care of ReLU activation outputs ?
*  How well does BBQ perform on architectures beyond transformers, such as CNNs or Vision transformers?
* Can the authors provide results isolating the contribution of each step (Hadamard Transform, Φ, RMS normalization, etc.) to the final accuracy and entropy gain?
* Does BBQ introduce any gradient instability during early training, especially in extremely low-precision (1–2 bit) setups?
* Could the authors discuss whether the binary search for Φ−1 values introduces any measurable latency overhead in hardware implementations?

---

> ### Author Response · Authors · 2025-11-22
> **Part 1 of 3**
>
> We thank Review BCbT for reviewing our paper and for providing extensive feedback. We are glad you found our quantization framework technically sound and conceptually novel.
>
> ### 1. Improving the Experimentation Section
>
> > Experiments are conducted exclusively on LLaMA-based architectures; it remains unclear whether BBQ generalizes to convolutional or vision transformer models.
>
> > How well does BBQ perform on architectures beyond transformers, such as CNNs or Vision transformers?
>
> > No comparison to information-theoretic quantizers beyond LSQ/QuEST: Including recent ITO-based methods (e.g., NF4, FP4, or adaptive quantizers) would strengthen the empirical claim of superiority.
>
> **Response** We have expanded our experimentation section to include results for models with another architecture, GPT, in addition to LLaMA. The GPT results are in Table 4. We have added comparison to quantizers beyond LSQ and QuEST, including ParetoQ-SEQ and NF4 as you suggested. These comparisons are also shown in Table 4. Results show BBQ consistently outperforms, by a large margin (up to 24 perplexity points), existing quantization methods on LLaMA (Table 3) and GPT (Table 4). While BBQ is designed for language models, we discuss in Section A.5, a variant of BBQ designed for vision models. Preliminary results on small models (5M to 20M parameters) show this BBQ variant can outperform QuEST and LSQ.
>
> **Action 1.1** We added QAPT results on the GPT architecture in Table 4, in addition to the LLaMA results in Table 3.
>
> **Action 1.2** We compare against NF4 in Table 4.
>
> **Action 1.3** We added preliminary results on small-scale vision models (ViT and Resnet) in Section A.5 and Table 10.
>
> ### 2. Ablation Analysis
>
> > Lack of ablation analysis: The impact of individual steps (e.g., Hadamard Transform, Φ smoothness, or γ initialization) is not quantitatively analyzed, leaving unclear which components are most critical.
>
> > Can the authors provide results isolating the contribution of each step (Hadamard Transform, Φ, RMS normalization, etc.) to the final accuracy and entropy gain?
>
> **Response** We added an ablation study in Table 5 and Section 4.3 covering/isolating the contribution of each of the five features of BBQ: Hadamard Transform, RMS normalization, the Gaussian CDF, gamma initialization, and gamma learnability. Results show the main perplexity reduction and entropy increase of BBQ comes from the combination of the Gaussian CDF and gamma initialization, whereas making gamma learnable can further reduce the perplexity by a small margin. Lastly, the features inherited from QuEST (Hadamard Transform and RMS normalization) also improves performance. For clarification, we also added Section A.4 and Figure 7 to illustrate the conceptual difference between BBQ and QuEST.
>
> **Action 2.1** We added ablation analysis in Table 5 and Section 4.3.
>
> **Action 2.2** We illustrate the conceptual difference between BBQ and QuEST in Section A.4 and Figure 7.
>
> ### 3. Inference speedup and different GPU generations
>
> > Implementation details underexplored: Although latency improvements are discussed, comparisons with more hardware-optimized kernels/different GPU generations or newer architectures like NPUs could provide stronger evidence of scalability.
>
> > Could the authors discuss whether the binary search for Φ−1 values introduces any measurable latency overhead in hardware implementations?
>
> **Response** We implemented BBQ inference on two generations of NVIDIA GPUs and with two different data types. Specifically, one of the GPUs is RTX 5090, an NVIDIA Blackwell GPU with FP4 tensor cores. The other GPU is A100 80GB, an NVIDIA Ampere GPU with int4 tensor cores. We test the end-to-end latency improvements on both hardware and present our results in Figure 6, which shows BBQ can improve inference latency by a factor of 40% regardless of hardware architecture / GPU generation. In addition, we show latency of the BBQ quantization kernel (binary search for Phi^{-1} values) as a fraction of the total latency in Figure 6, which shows the overhead is negligible compared to the latency savings on matrix multiplication.
>
> **Action 3.1** We implemented BBQ inference on two generations of NVIDIA GPUs and included our inference speedup profiling results in Figure 6. Figure 6 also illustrates the latency of binary search is negligible compared to the speedup it brings.

---

> > ### Comment · Reviewer_BCbT · 2025-11-22
> > **Thank you for your answers and updated sections**
> >
> > Appreciate Authors for providing the answers to questions as well as updated sections as needed. No further questions on this.

---

> > > ### Author Response · Authors · 2025-11-24
> > >
> > > We once again thank Reviewer BCbT for providing concrete suggestions to improve our paper.

---

> ### Author Response · Authors · 2025-11-22
> **Part 2 of 3**
>
> ### 4. Clarification
> > section 2.3, "learning is domain agnostic" : the two example to support this is very simplistic assumption. For eg. rotated, cropped and color jitter image is still image and in the same domain, images transformed to frequency domain is still an image but encoded in different format. In both cases, domain is still image processing. Autoencoder is a bit closer but they can't be used to train cross domain models. Authors are suggested to add more concrete examples and possibly quantification of this assumption Limited scope of evaluation:
>
> **Response** We apologize for the miscommunication/misunderstanding. By “domains”, we do not mean “modality” as in text vs. images. Instead, we are referring to different “units” or “formats” of the same underlying information. While an image remains an image (its underlying information is the same), applying a discrete fourier transform/discrete cosine transform converts the image from the “spatial domain”, where the units are intensity at a location, to the “frequency domain”, where the units are intensity at a frequency. As another example, while a measurement of temperature may reside in the “Celsius domain”, the function f(x)=9*x/5 + 32 can convert it into the “Fahrenheit” domain, despite the underlying abstract notion of temperature being the same. Similarly, an encoder side of an autoencoder transforms the data into a latent compact space, while the decoder side applies a (pseudo-)inverse transform to convert the data back to the original domain. By “learning is domain agnostic”, we mean neural networks can learn from temperature measurements regardless of whether the specific unit used is Celsius or Fahrenheit, since NNs are universal approximators.
>
> However, as we mentioned in Section 2.3, an important caveat is “information is preserved” during the conversion process. If our conversion function is lossy and corrupts the underlying information by adding too much noise, then most information of the underlying data is lost and the converted data becomes dominated by pure noise. BBQ is designed to maximally preserve information given its limited precision.
>
> In addition, we acknowledge the domain-agnostic property of learning has its limitations, as real networks are susceptible to gradient explosion/vanishing. Therefore, transforming the data into a domain where each data point has very large magnitude can often lead to gradient explosion. As such, normalization is typically applied to data before feeding them into neural networks to regulate the magnitude of data points. BBQ is also designed to maintain this property. Instead of transforming the data into arbitrary output domains, we specifically choose a domain where data points have similar magnitude to the input domain. Our ablation in Table 5 shows gamma initialization (which controls the magnitude of data points in the output domain) plays a significant role in achieving good performance.
>
> > Since BBQ operates in separate input/output domains and does not minimize reconstruction error, it may not perform well in fine-tuning or post-training quantization scenarios — limiting its versatility.
>
> Specialization is a well-known technique to overcome the limitations of general-purpose methods. For example, by specializing in software with abundant parallelism, GPUs can significantly outperform CPUs, even though CPUs can technically run more types of software than GPUs. In the field of DNN quantization, methods like PV-Tuning [1] and QuZO [2] specialize in quantization-aware fine-tuning (QAFT), while methods like FOGZO [3] specializes in quantization-aware pre-training (QAPT).
> By removing the constraint that a quantizer must apply to both PTQ, QAFT, and QAPT, BBQ can specialize to QAPT and utilize the domain-agnostic property of learning to maximally preserve information while remaining compute-efficient. In other words, while we have explicitly acknowledged this limitation in the original submission, we additionally argue BBQ is specialization by design, and its weakness is what enables its strengths.
>
> [1] PV-Tuning: Beyond Straight-Through Estimation for Extreme LLM Compression
>
> [2] QuZO: Quantized Zeroth-Order Fine-Tuning for Large Language Models
>
> [3] Improving the Straight-Through Estimator with Zeroth-Order Information

---

> ### Author Response · Authors · 2025-11-22
> **Part 3 of 3**
>
> ### 4. Clarification (Continued)
> > section 3.4 : Uniform quantization and unsigned to signed conversion : Does it make any difference for ReLU based models where output activations are always positive and unsigned values are preferred for better accuracy. ? I support this sentence "we subsequently convert the positive on data to symmetric data by subtracting 2b−1 and the hyperparameter zero point z," is to take care of ReLU activation outputs ?
>
> We do not notice any difference in ReLU-based models, nor do we give ReLU networks any special treatments. Even in ReLU/GELU based models, the hadamard transform still converts the post-ReLU activations to be empirically symmetric, since the Hadamard transform is a Central-Limit-Theorem-based Gaussianization operator. In other words, regardless of the symmetry of activations before the hadamard-transform, they remain approximately symmetric after the hadamard transform.
>
> While the Phi function functionality clips the data into a finite range, it takes a symmetric input and returns outputs within the range (0, 1). Effectively, Phi effectively converts the post-Hadamard data from symmetric to asymmetric. The subsequent shifting is to restore this symmetric/zero-mean property. Our results in Section A.5 shows BBQ outperforms LSQ on GeLU networks, despite the fact that LSQ does keep the asymmetric nature of activations by clipping within the range [0, 2^p - 1] instead of between the range [-2^{p-1}, 2^{p-1}-1].
>
> > Does BBQ introduce any gradient instability during early training, especially in extremely low-precision (1–2 bit) setups?
>
> BBQ does not introduce any gradient instability during early training in any case. As shown in the training loss curves in Section A.6, the loss of BBQ consistently reduces at a steady pace and does not suddenly increase over a short period (overshooting/divergence).

---

### Official Review · Reviewer_qRhd · 2025-10-31

**Soundness:** 2
**Presentation:** 2
**Contribution:** 2
**Rating:** 4
**Confidence:** 5

**Summary:**

This paper introduces BBQ (Bell Box Quantization), an information-theoretically optimal (ITO) quantization method for quantization-aware pre-training (QAPT) of large language models. BBQ builds on prior work, such as QuEST and NormalFloat, by introducing a Gaussian CDF on the normalised Hadamard transform before applying a learnable bit-dependent scaling, resulting in a quantizer that is both ITO- and compute-efficient. The key idea is that quantization can occur in one domain while producing outputs in a different, hardware-efficient domain, thus preserving information content without losing computational advantages.

The authors conduct experiments on LLaMA models of various sizes, demonstrating improvements in quantized weight entropy and C4 perplexity over QuEST and LSQ. The proposed quantizer also supports efficient implementations on emerging low-precision formats, such as MXFP4.

**Strengths:**

The authors correctly note that existing QAPT methods are limited in their representation capacity, as they implicitly constrain the entropy of the quantised weights. They ground this hypothesis clearly and use it to motivate the development of the first information-theoretically optimal (ITO) quantizer for QAPT. The resulting method, BBQ, is conceptually straightforward yet effective, consistently improving over strong baselines such as QuEST and LSQ across multiple bit widths.

A key strength of the paper is its demonstration that introducing the Gaussian CDF transformation to achieve ITO quantization does not incur additional computational overhead, owing to an efficient implementation that maintains compatibility with low-precision arithmetic.
It is also interesting to see how the authors explore and integrate emerging numerical formats, such as MXFP, highlighting the method’s forward-looking relevance to upcoming GPU architectures.

**Weaknesses:**

In general, the premise of the paper is sound but requires significant work in presentation, ablations and experimental results to qualify for the conference:

* The experimentation section is lacking. I would like to see zero-shot results and one more family of models, rather than just increasing the size of the same architecture. Please include a comparison to other SOTA methods, such as ParetoQ.

* From a methodological standpoint, the contribution feels incremental relative to QuEST: most of the quantization pipeline (Hadamard transform, normalisation, uniform rounding) is directly inherited, and the main novelty — applying a Gaussian CDF and learning a scaling factor — would benefit from deeper ablations against QuEST’s corresponding components. This would make it more straightforward to determine how much of the gain comes from the CDF transformation itself.

* More results and information are required regarding inference speed-up (see questions below)

* The presentation of the paper has issues (see details in the question below).

**Questions:**

* Table 1: How is MXFP4 integrated into the method, given that MXFP4 is a dynamic quantisation scheme with an E8M0 scale and FP4 data?
* line 281: A smoother function can be: any experiments or reference to support this statement?
* Table 3: Please provide a description of the heads in the caption
* Table 3: It would be easier to read the table if the precision were presented as the difference from the full-precision
* The argument that quantised weight entropy is a good proxy for the final ppl is fundamental to the results section. However, there are no ablation studies or experiments to validate this hypothesis.
* Please add legends to all figures with training curves, cause they are hard to read.
* Section 4.2 requires more work: the latency overhead of the quantization function should be given as a fraction of the total latency. I would recommend showing the median latency over several runs for the full LLama models and using profiling to illustrate the proportion of the quantization kernel. In addition, it would be nice to quantify the total speed-up of the method compared to NormalFloat and then the full-precision baseline in a table
* A competing QAT method: Learnable Companding Quantization for Accurate Low-bit Neural Networks by Kohei Yamamoto introduces a similar concept where the transformation function is learnt. Although it is an older paper, it follows a similar logic. Have yοu considered comparing to that method too? They have awe-inspiring results for ResNets
* Appendix A2: From the main text, I understood that EMA of the reciprocal of $\sigma$ is part of the method. What do you compare against here?
* Appendix A3: I don’t think that adding all these validation losses adds much value to the paper. Expanding your results section to include more architectures and zero-shot results would be more impactful

---

> ### Author Response · Authors · 2025-11-22
> **Part 1 of 3**
>
> We thank Review qRhd for reviewing our paper and for providing extensive feedback. We are glad you found the premise of our paper sound.
>
> ### 1. Improving the Experimentation Section
> > The experimentation section is lacking. I would like to see zero-shot results and one more family of models, rather than just increasing the size of the same architecture. Please include a comparison to other SOTA methods, such as ParetoQ.
>
> **Response**: we have expanded the experimentation section to additionally include results of the models in the GPT family, instead of just having the models with the LLaMA architecture. Results (in Table 4) show BBQ consistently outperforms other SOTA methods on GPT models. We added zero-shot results in Section A.3 and Table 9. We added comparison to other SOTA methods such as ParetoQ in Table 4. In addition, we note that ParetoQ uses LSQ (without any modification) for 3-bit and 4-bit quantization, and proposes its own novel technique SEQ for 2-bit quantization. Since we already have comparisons against LSQ, we added comparisons against SEQ for 2-bit quantization.
>
> As a side note, ParetoQ is published at NeurIPS 2025, whose publishing date is one month after the ICLR submission deadline. According to the ICLR 2026 policies posted here  https://iclr.cc/Conferences/2026/ReviewerGuide, we believe ParetoQ should be considered contemporaneous.
>
> **Action 1.1** We added QAPT results on the GPT architecture in Table 4, in addition to the LLaMA results in Table 3.
>
> **Action 1.2** We compare against the SOTA method Pareto-SEQ in Table 4.
>
> **Action 1.3** We added zero-shot results In Section A.3 and Table 9.
>
> ### 2. Deeper Ablation
> > From a methodological standpoint, the contribution feels incremental relative to QuEST: most of the quantization pipeline (Hadamard transform, normalisation, uniform rounding) is directly inherited, and the main novelty — applying a Gaussian CDF and learning a scaling factor — would benefit from deeper ablations against QuEST’s corresponding components. This would make it more straightforward to determine how much of the gain comes from the CDF transformation itself.
>
> **Response** We have added an ablation study in Table 5 and Section 4.3 showing that the Gaussian CDF (probability integral transform) plus the initialization of the scaling factor contributes to most of the gain, whereas making the scaling factor learnable (instead of leaving it as a constant) can further reduce perplexity by a smaller margin. In addition, Table 5 also shows features inherited from QuEST such as the Hadamard transform and RMS normalization can improve performance. We have also added Section A.4 and Figure 7 to illustrate and explain the conceptual difference between BBQ and QuEST.
>
> **Action 2.1** We added an ablation study in Table 5 and Section 4.3.
>
> **Action 2.2** We added Section A.4 and Figure 7 to illustrate the conceptual difference between BBQ and QuEST.
>
>
> ### 3. More information on Inference speedup
> > More results and information are required regarding inference speed-up
>
> > Section 4.2 requires more work: the latency overhead of the quantization function should be given as a fraction of the total latency. I would recommend showing the median latency over several runs for the full LLama models and using profiling to illustrate the proportion of the quantization kernel. In addition, it would be nice to quantify the total speed-up of the method compared to NormalFloat and then the full-precision baseline in a table
>
> **Response** Thank you for the great advice. We used profiling tools to measure the end-to-end LLaMA inference latency on two modern NVIDIA GPUs (RTX 5090 and A100 80GB) and showed our profiling results in Figure 6. The latency overhead of the quantization function is illustrated as the green regions in Figure 6. Following your advice, Figure 6 quantifies and illustrates the speed-up of the method compared to NormalFloat and the full-precision baseline. We have also discussed Figure 6 in Section 4.2. BBQ is 40% faster than full-precision baseline and 48% faster than NormalFloat.
>
> **Action 3.1** We added Figure 6 to illustrate our profiled end-to-end LLaMA inference latency breakdown, quantization latency overhead, and speedup over full-precision and NormalFloat.

---

> ### Author Response · Authors · 2025-11-22
> **Part 2 of 3**
>
> ### 4. Improving the Presentation
>
> > Table 3: Please provide a description of the heads in the caption
>
> > Table 3: It would be easier to read the table if the precision were presented as the difference from the full-precision
>
> > Please add legends to all figures with training curves, cause they are hard to read.
>
> **Response**  We added descriptions of the headers in the caption of Table 3. We added legends to all figures with training curves, which are now in Section A.6. We have a question regarding your second suggestion (see below)
>
> **Action 4.1** We added descriptions of the headers in the caption of Table 3.
>
> **Action 4.2** We have added legends to all figures with training curves in Section A.6.
>
> **Question 4.1** By “the difference from the full-precision”, do you mean a 3-bit model (precision = 3 bits) should be presented as “-13 bits” in Table 3, since full-precision is 16 bits?
>
> ### 5. Clarification
> > Table 1: How is MXFP4 integrated into the method, given that MXFP4 is a dynamic quantisation scheme with an E8M0 scale and FP4 data?
>
> During inference, our BBQ quantization method generates a quantized activation matrix qx.
>
> We then launch a low-precision matrix multiplication kernel to compute the matrix product of qx and the offline-pre-computed quantized weight matrix qw. On blackwell architectures, we use torch._scaled_mm to launch an FP4 matrix multiplication kernel. However, the API requires us to include two arrays (one for weights, one for activations) of FP8 scales as you suggested. For simplicity, we pass in two arrays of ones (each element in the arrays is set to the number 1.0 expressed in FP8). On Ampere architectures, we use cutlass to launch an INT4 matrix multiplication kernel.
>
> Lastly, we launch an element-wise multiplication kernel to scale the matrix product of qx and qw by a factor of sx*sw, which is also pre-computed offline. As future work, it may be possible to incorporate sx and sw into the FP8 scale factor arrays and avoid the overhead of the last element-wise multiplication kernel altogether. Despite missing this opportunity of optimization, our profiling results shows we can still achieve 40% speedup over the full-precision baseline.
>
> > line 281: A smoother function can be: any experiments or reference to support this statement?
>
> We have edited the paper and added two citations [1][2] to support this statement. Gradient-based methods generally rely on some notion of smoothness to guarantee stable updates and convergence.
>
> [1] Yurii Nesterov. Introductory Lectures on Convex Optimization: A Basic Course.
>
> [2] Leon Bottou, Frank E. Curtis, and Jorge Nocedal. Optimization methods for large-scale machine learning, 2018
>
> > The argument that quantised weight entropy is a good proxy for the final ppl is fundamental to the results section. However, there are no ablation studies or experiments to validate this hypothesis.
>
> **Response** We derive this argument based on multiple related works. Quantization entropy correlates with model accuracy has been empirically confirmed in prior works [1][2]. In addition, it is known in information theory that the empirically measured entropy is an upper bound to the amount of information in the data [3]. Since the next-token prediction training objective explicitly encourages the model to memorize its training set, LLMs need to have the ability to store sufficient information (either by having many parameters, or high precision, or both). If LLMs don’t store much information, they cannot memorize well. The precision scaling law [4] empirically confirms this: a model’s perplexity can be accurately predicted from its parameter count and precision, which is an upper bound of its entropy. This is where we draw our inspiration: if we explicitly increase entropy, the upper bound of information, would the model be more capable in accumulating information from training data? Our experiment results show BBQ achieves lower perplexity and higher entropy than prior works, and we empirically observe a correlation between the two over multiple experiments.
>
> Lastly, we did ablation studies in Table 6 and Section 4.3, showing that the normal CDF (the key operator that increases entropy for Gaussian-like data) contributes to a perplexity decrease of 4 points on a 2-bit LLaMA-95M model.
>
> [1] Ecco: Improving memory bandwidth and capacity for llms via entropy-aware cache compression
>
> [2] Edgeqat: Entropy and distribution guided quantization-aware training for the acceleration of lightweight llms on the edge.
>
> [3] Information Theory: A Tutorial Introduction.
>
> [4] Scaling laws for precision.

---

> ### Author Response · Authors · 2025-11-22
> **Part 3 of 3**
>
> ### 5. Clarification (Continued)
>
> > A competing QAT method: Learnable Companding Quantization for Accurate Low-bit Neural Networks by Kohei Yamamoto introduces a similar concept where the transformation function is learnt. Although it is an older paper, it follows a similar logic. Have yοu considered comparing to that method too? They have awe-inspiring results for ResNets
>
> We find LCQ to be an elegant non-uniform quantization method, but BBQ is fundamentally different from LCQ.
>
> LCQ falls under the umbrella of “same-domain” quantizers, in that it uses the following formulation:
>
> $$\hat{x} = f^{-1}(q_x), q_x = r(f(x))$$
>
> $$\hat{w} = f^{-1}(q_w), q_x = r(f(w))$$
>
> Note that if the rounding/discretization function r produces very little rounding error, then f and f^{-1} should cancel out, making xhat and x (what and w) numerically equivalent (hence “same-domain”).
>
> Compute-efficient quantization methods use a linear f^{-1} and utilize the fact that
> $$x \cdot w = f^{-1}(f^{-1}(q_x \cdot q_w)) $$
> to first perform computation in low-precision and later dequantize the result to high-precision. However, LCQ uses a custom non-linear f^{-1}. As such, LCQ cannot first compute in low-precision and later dequantize to high-precision. In other words, while LCQ can reduce memory consumption, it does not improve compute efficiency.
>
> BBQ is designed to counter exactly this limitation of non-uniform quantizers. BBQ is a “cross-domain” quantizer, in that it uses the following formulation:
>
> $$\hat{x} = g(q_x), q_x = r(f(x))$$
>
> $$\hat{w} = g(q_w), q_w = r(f(w))$$
>
> where $$g \neq f^{-1}$$
>
> BBQ intentionally chooses an f to improve entropy, an r to produce a compute efficient data type (int4/fp4), and a linear g to improve compute efficiency. Since g != f^{-1}, xhat does not live in the same domain as x. This is the key characteristic that separates BBQ from same-domain quantizers, and what enables BBQ to be both ITO and compute-efficient.
>
> Lastly, while we would like to compare against LCQ, we could not find its source code and hence cannot perform an apples-apples comparison. In our revised paper uploaded above we have cited LCQ.
>
> > Appendix A2: From the main text, I understood that EMA of the reciprocal of σ is part of the method. What do you compare against here?
>
> During training, BBQ always uses the reciprocal of σ, but its inference behaviour is different depending on the BBQ variant. Variant 1 uses the EMA of the reciprocal of σ measured during training, denoted as E[1/σ], at inference time. Variant 2 shares the same behaviour as training, in that it measures reciprocal of σ directly from the inference activations. Here we are comparing the two variants of BBQ and show that they achieve identical performance. The goal is to verify that the use of EMA statistics does not cause a mismatch between training and inference, and degrade accuracy.
>
> > Appendix A3: I don’t think that adding all these validation losses adds much value to the paper. Expanding your results section to include more architectures and zero-shot results would be more impactful.
>
> As discussed above, we have significantly expanded our results section.

---

### Official Review · Reviewer_24io · 2025-11-02

**Soundness:** 3
**Presentation:** 3
**Contribution:** 3
**Rating:** 8
**Confidence:** 3

**Summary:**

The paper proposes to improve LLM quantized training by transforming the input to enable desired properties (entropy) in the network layers.  The quantization method aims to utilize all of the quantized levels equally often. The input quantization consists of seven steps and includes Hadamard transform, RMS normalization, probability integral transform, and so on. Even though no formal proof of equal utilization has been provided, empirical evidence shows that entropy increases compared to SOTA. Empirical evidence also indicates that perplexity is decreased, resulting in improved performance.

**Strengths:**

1. The paper proposes to transform the input for improved performance. This idea is novel and does lead to better-performing LLMs.

**Weaknesses:**

1. It is not known if just transforming the input would also lead to deeper layers, also preserving the "utilize all of the quantized levels equally often" property. Though the transformer layer norm would bias it to do just that. So it's not that big of a problem.

**Questions:**

1. Overall, the paper is acceptable to me. I am curious to know if there is a straightforward way to achieve something similar with non-transformer models, such as CNNs.

---

> ### Author Response · Authors · 2025-11-22
>
> We thank Review 24io for reviewing our paper. We are glad you found our idea novel.
>
> ### 1. Clarification
> > It is not known if just transforming the input would also lead to deeper layers, also preserving the "utilize all of the quantized levels equally often" property. Though the transformer layer norm would bias it to do just that. So it's not that big of a problem.
>
> **Response** Activations in deep neural networks tend to follow bell-shaped distributions regardless of architecture [1]. On top of this, as discussed in Section 3.1 and 3.3, BBQ uses the Hadamard transform which can further encourage Gaussianity [2]. Assuming the post-Hadamard-transform activation closely matches a Gaussian, BBQ subsequently applies the probability integral transform/Gaussian CDF to create a likely uniformly distributed activation. The uniform distribution is known to maximize entropy provided data has a finite range.
>
> [1] Improving Neural Network Quantization without Retraining using Outlier Channel Splitting
>
> [2] QuEST: Stable Training of LLMs with 1-Bit Weights and Activations
>
> ### 2. CNNs
> > Overall, the paper is acceptable to me. I am curious to know if there is a straightforward way to achieve something similar with non-transformer models, such as CNNs.
>
> **Response** While BBQ is designed for language models, we discuss in Section A.5, a variant of BBQ designed for vision models such as CNNs/ResNets and DeiTs. Preliminary results on small models (5M to 20M parameters) show this BBQ variant can outperform QuEST and LSQ.
>
> **Action 2.1** We have added evaluation of a BBQ variant on vision models in Section A.5.

---

### Meta-Review · Area_Chair_krxp · 2026-01-01

**Summary:**

This paper introduces **BBQ (Bell Box Quantization)**, a **quantization-aware pre-training (QAPT)** method that aims to be both **information-theoretically optimal (ITO)** and **compute-efficient**. The key idea is to perform ITO quantization in the input domain while **outputting in a hardware-friendly domain** (e.g., INT/FP formats), enabling higher entropy/capacity utilization without sacrificing efficient compute.

Across **1–4 bit** training, the authors report consistent **perplexity improvements** over strong baselines (e.g., QuEST/LSQ and added comparisons in the rebuttal), and support practicality with **profiling-based end-to-end inference speedups** and expanded ablations identifying the main drivers of gains (notably the Gaussian CDF transform and scaling initialization). The rebuttal directly addresses the main reviewer concerns by adding broader experiments (including GPT and preliminary vision results), deeper ablations, and clearer latency breakdowns.

I recommend **Accept (poster)**: the core contribution is novel and practically motivated, the empirical evidence is strong after the rebuttal additions, and remaining issues are primarily presentation polish rather than technical gaps.

**Reviewer Concerns:**

* **Limited experimental diversity (tested only on LLaMA / single model family)** *(qRhd, VA2y, BCbT)*: **Addressed.** The rebuttal adds results on **GPT-family models** (beyond LLaMA), expands baseline coverage (e.g., ParetoQ/SEQ, NF4), and adds **zero-shot** evaluation.

* **Need broader model/setting coverage (zero-shot; more than scaling the same architecture)** *(qRhd)*: **Addressed.** The authors add **zero-shot results** and broaden evaluation to **GPT** in addition to scaling LLaMA sizes.

* **Generalization beyond Transformers (CNNs / vision models)** *(24io, BCbT, qRhd)*: **Partially addressed.** The rebuttal adds a **vision-oriented BBQ variant** with **preliminary** results on small vision models (ViT/ResNet) and discusses how to adapt BBQ.
  **Still outstanding:** evidence on large-scale vision settings remains limited; the strongest validation is still in LLM/QAPT.

* **Method feels incremental relative to QuEST; unclear what drives gains** *(qRhd)*: **Addressed.** The rebuttal adds **deeper ablations** against QuEST-style components and clarifies conceptual differences (including new explanatory sections/figures).

* **Lack of ablation analysis across pipeline steps (Hadamard/RMS/CDF/γ init/γ learnability)** *(qRhd, BCbT)*: **Addressed.** The rebuttal adds a step-wise **ablation table** isolating each component and showing which contributes most.

* **Inference speed / deployment practicality: need end-to-end profiling; quantify quantization overhead (Φ⁻¹ / binary search)** *(qRhd, BCbT)*: **Addressed.** The rebuttal adds **profiling-based latency breakdowns** (including quantization kernel overhead as a fraction of total latency) and reports end-to-end speedups over FP and NormalFloat on multiple GPUs.

* **Request for more hardware/format clarity (e.g., MXFP4 integration; smoother function claim)** *(qRhd)*: **Addressed.** The authors clarify how FP4/MXFP-style kernels are used in practice and add citations/support for optimization statements.

* **Presentation / readability issues (tables harder to parse; missing legends; unclear headers/captions)** *(qRhd)*: **Addressed.** The rebuttal adds **header descriptions**, improves captions, and adds **legends** to training-curve figures.

* **Justification that entropy is a good proxy for perplexity / quality** *(qRhd, BCbT, VA2y)*: **Mostly addressed.** The authors add citations plus ablation evidence (e.g., showing the CDF step increases entropy and reduces PPL).
  **Still somewhat outstanding:** the argument remains largely **empirical/correlational** rather than a formal guarantee, though that may be acceptable for an empirical QAPT paper.

* **“Learning is domain agnostic” motivation is unclear / too simplistic** *(BCbT)*: **Addressed.** The authors clarify “domain” as **units/representation** (not modality), add concrete examples (spatial↔frequency; Celsius↔Fahrenheit), and discuss stability caveats (normalization/magnitude control).

* **Versatility concern: cross-domain quantizer may not suit PTQ / fine-tuning settings** *(BCbT)*: **Addressed as a scope clarification.** The authors position BBQ as **specialized for QAPT by design**, explicitly acknowledging it may not be a general PTQ/QAFT method.

* **Comparison to alternative “learned transform” quantizers (e.g., LCQ)** *(qRhd)*: **Partially addressed.** The rebuttal explains conceptual differences and cites LCQ.
  **Still outstanding:** no direct experimental comparison is provided (authors note lack of code).

* **Scalability / practicality for larger parameter scales (≥300M, 1B+) and training cost** *(VA2y)*: **Partially addressed.** The rebuttal argues BBQ has similar pretraining cost to other QAPT methods and cites inference speedups, but cannot run much larger-scale pretraining due to compute limits.
  **Still outstanding:** limited **direct empirical** evidence at >300M–1B+.

* **Stability at extreme low-bit (1–2 bit); divergence concerns; LSQ limitations** *(VA2y, BCbT)*: **Addressed.** The authors state BBQ does not show early-training instability, point to training curves, and clarify BBQ supports **1-bit** regimes where LSQ diverges.

* **ReLU/unsigned vs signed conversion detail** *(BCbT)*: **Addressed.** The rebuttal explains why symmetry is restored after Φ (and why Hadamard tends to Gaussianize/symmetrize activations even after ReLU/GELU).

**Reviewer Scores:**

## Reviewer 24io: 8 → 8
Why: Already strongly positive (“accept, good paper (poster)”) with only a curiosity about non-transformer applicability. The authors added a vision-variant discussion and preliminary vision results, which likely reinforces the existing score rather than changing it.

## Reviewer qRhd: 4 → 6
Why: The initial score was driven by (i) limited experiments (only LLaMA, missing zero-shot), (ii) insufficient ablations vs QuEST, and (iii) missing end-to-end inference profiling/speedup evidence and presentation issues. The rebuttal directly addresses these: adds GPT + zero-shot + ParetoQ/NF4 comparisons, adds step-wise ablations isolating the CDF/scaling contributions, and includes profiling-based latency breakdowns across GPU generations. With the main technical and empirical concerns addressed, a move from “marginally below threshold” to “weak accept” is plausible.

## Reviewer BCbT: 6 → 6
Why: This reviewer was already at “marginally above threshold” and, after the authors’ responses and updates, explicitly replied that they had **no further questions**. Given that stance, the most likely outcome is that the score remains unchanged (already consistent with acceptance), rather than increasing further.

## Reviewer VA2y: 4 → 6
Why: Their main concerns were (i) generalization beyond LLaMA and (ii) practicality/viability. The rebuttal adds GPT results (addressing generalization), clarifies that BBQ supports 1-bit settings where LSQ fails, and reinforces practicality with profiling showing substantial inference speedups. While the work remains an empirical QAPT study and cannot demonstrate very large-scale (>300M–1B+) pretraining due to compute limits, the added breadth and deployment evidence are sufficient to move from “marginally below threshold” to a **marginally above threshold**,  leaning toward acceptance.

---

### Decision · Program_Chairs · 2026-01-26

Accept (Poster)